# On the Entropy Dynamics in Reinforcement Fine-Tuning of Large Language Models

## Abstract

Entropy serves as a critical metric for measuring the diversity of outputs generated by large language models (LLMs), providing valuable insights into their exploration capabilities. While recent studies increasingly focus on monitoring and adjusting entropy to better balance exploration and exploitation in reinforcement fine-tuning (RFT), a principled understanding of entropy dynamics during this process is yet to be thoroughly investigated. In this paper, we establish a theoretical framework for analyzing the entropy dynamics during the RFT process, which begins with a discriminant expression that quantifies entropy change under a single logit update. This foundation enables the derivation of a first-order expression for entropy change, which can be further extended to the update formula of Group Relative Policy Optimization (GRPO). The corollaries and insights drawn from the theoretical analysis inspire the design of entropy controlling methods, and also offer a unified lens for interpreting various entropy-based methods in existing studies. We provide empirical evidence to support the main conclusions of our analysis and demonstrate the effectiveness of the derived entropy-discriminator clipping methods. This study yields novel insights into RFT training dynamics, providing theoretical support and practical strategies for optimizing the exploration-exploitation balance during LLM fine-tuning.

## 1 Introduction

Reinforcement fine-tuning (RFT) (OpenAI, 2025) has recently attracted growing attention as a post-training paradigm for enhancing the capabilities of large language models (LLMs) (Guo et al., 2025; Yang et al., 2025a; Agarwal et al., 2025). It has shown substantial improvements across a range of downstream tasks, such as mathematical reasoning (Shao et al., 2024; Chen et al., 2025), programming (Wei et al., 2025; Zeng et al., 2025), and tool usage (Zhang et al., 2025; Feng et al., 2025).

Drawing from reinforcement learning (RL) (Sutton et al., 1998), RFT transforms the fine-tuning process into a policy optimization problem where LLMs are incentivized to produce high-reward responses. The exploration-exploitation trade-off presents a crucial challenge for RFT, potentially leading to unstable performance and stagnation in local optima (Arulkumaran et al., 2017; Ahmed et al., 2019). In this context, the *entropy* of responses emerges as a key diagnostic metric, offering insights into the output diversity of LLMs, and is actively leveraged by recent studies (Yu et al., 2025; Cui et al., 2025; Hu et al., 2025) to monitor training dynamics and regulate policy behavior.

However, existing methods (Wang et al., 2025; Liao et al., 2025; Yu et al., 2025; He et al., 2025) often rely on heuristic designs that treat entropy in isolation and oversimplify its adjustment. Moreover, the divergence in whether these approaches encourage or suppress entropy highlights a fundamental lack of in-depth understanding of entropy dynamics (Hu et al., 2025; Luo et al., 2025; An et al., 2025). Such an unprincipled basis can lead to labor-intensive hyperparameter tuning without clear guidance, thus hindering the effective optimization of RFT. As a result, a theoretically grounded framework is increasingly necessary to characterize entropy dynamics in RFT.

To fulfill this gap, we establish a theoretical framework that provides a principled understanding of entropy dynamics in RFT. Inspired by (Ren & Sutherland, 2025), we model the update of a single token's logit during optimization, and characterize how it propagates through the model's output probability distribution, ultimately influencing the policy's entropy. Our derivation reveals that the entropy change direction is determined by the interplay between the update direction (whether the

token is rewarded or penalized) and the sign of the proposed discriminator score $S_*$, which captures the relationship between token probability and policy entropy. This analysis explains the widely observed phenomenon of rapid entropy collapse (Liao et al., 2025; Yu et al., 2025) when models are consistently rewarded for generating high-probability and "safe" responses (He et al., 2025).

Building upon such single-token analysis, we extend our framework to analyze the entropy change resulting from an optimization step under Group Relative Policy Optimization (GRPO) (Shao et al., 2024). We derive an expression that practically computes the entropy change trend leveraging the discriminant and it's policy-weighted expectation. Our analysis provides insights for the development of entropy-based methods, both inspiring practical clipping strategies and shedding light on the mechanisms of existing approaches.

Our contributions can be summarized as follows:

- We propose a theoretical framework that characterizes the token-level entropy change during policy optimization. We further extend it to a practical GRPO optimization step and derive a first-order analytical expression, indicating that the direction of entropy change is closely related to the direction of token updates and a discriminator score $S_*$.

- Our theoretical analysis provides new insights for the design of entropy controlling methods. Upon this, we explain existing entropy controlling methods from the perspective of entropy dynamics, offering a unified and principled theoretical framework for understanding their effects and underlying mechanics.

- We conduct experiments to provide empirical evidence for our theoretical analysis, showing that $S_*$ can be a reliable discriminator for the entropy dynamics. The experimental results also demonstrate the effectiveness of the derived entropy-discriminator clipping methods in stabilizing the entropy in RFT to promote model exploration.

## 2 PRELIMINARIES

**Group Relative Policy Optimization (GRPO)** GRPO (Shao et al., 2024) is a prominent RFT algorithm that has proven highly effective and efficient across various tasks (Guo et al., 2025; Zhang et al., 2025). In GRPO, for each query $q$, a behavior policy $\pi_{\theta_{\text{sample}}}$ is employed to sample a group of $G$ responses $\{o_i\}_{i=1}^G$, where each response $o_i = (a_{i,1}, \ldots, a_{i,T_i})$ with $T_i$ tokens is subsequently assigned a scalar reward $R_i$, and each token $a_{i,t}$ in response is generated under state $s_{i,t} = (q, o_{i,<t})$. The policy is updated by maximizing the GRPO objective function, defined as:

$$\mathcal{J}_{\text{group}}(\theta) = \frac{1}{\sum_{j=1}^G T_j} \sum_{i=1}^G \sum_{t=1}^{T_i} \min\Big(r_{i,t}(\theta)\, A_i,\, \text{clip}\big(r_{i,t}(\theta),\, 1-\varepsilon,\, 1+\varepsilon\big)\, A_i\Big). \tag{1}$$

Following (Yu et al., 2025; An et al., 2025; Wang et al., 2025), we omit the KL divergence penalty. Here, the advantage $A_i$ is computed by standardizing rewards within the group, and the importance ratio $r_{i,t}(\theta)$ is the token-level probability ratio between the target and behavior policies, i.e., $A_i = \frac{R_i - \text{mean}(\{R_j\}_{j=1}^G)}{\text{std}(\{R_j\}_{j=1}^G)}$ and $r_{i,t}(\theta) = \frac{\pi_\theta(a_{i,t}|s_{i,t})}{\pi_{\theta_{\text{sample}}}(a_{i,t}|s_{i,t})}$. The parameter $\varepsilon$ defines the clipping range for PPO-style (Schulman et al., 2017) clipped objective. In our "strict on-policy training" (Chen et al., 2025) setup, where the behavior policy is the optimized policy ($\pi_{\theta_{\text{sample}}} = \pi_\theta$), the importance ratio satisfies $r_{i,t} = 1$ and the clipping mechanism remains inactive. In this case, the update of GRPO encourages increasing the probability of sampled tokens if $A_i > 0$ and decreasing it if $A_i < 0$.

**Entropy Dynamics** Entropy provides a principled measure of uncertainty in a probability distribution and is used to quantify the diversity of model outputs. For an LLM, the next-token distribution is given by $\mathbf{p}_t(\cdot) = \pi_\theta(\cdot \mid s_t) = \text{softmax}(\mathbf{z}_t)$, where $\mathbf{z}_t$ are the model's logits at position $t$ in a response. The token-level entropy is then defined as $H_t = -\sum_{i \in [V]} p_i^t \log p_i^t$, where $V$ denotes the size of vocabulary $\mathcal{V}$ and $p_i^t$ is the probability of token $a_t$ being the $i$-th vocabulary item.

The field of *learning dynamics* (Ren & Sutherland, 2025) studies how parameter updates affect model predictions. In this work, we introduce the concept of *Entropy Dynamics*, focusing specifically on how token entropy evolves during reinforcement fine-tuning (RFT). Specifically, we investigate how a parameter update, triggered by a single sampled token $a_t$, alters the entropy of the output token distribution at that step.

We formalize this by investigating the relationship between the entropy change before and after update, $\Delta H_t$, and the distribution of the policy in position t, $\pi_\theta(a_t \mid s_t)$. By analyzing this relationship, we aim to uncover the principles that determine whether the updates in RFT encourage diverse responses or lead to repetitive, similar outputs.

## 3 ANALYSIS OF THE ENTROPY DYNAMICS IN RFT

To establish a principled understanding of entropy dynamics in RFT, we propose a theoretical framework that characterizes the token-level entropy change during policy optimization. Specifically, we quantify how the update of a single token affects the policy's entropy, providing a microscopic view of entropy dynamics. Upon this, we derive the first-order expression for the entropy change resulting from a policy update step when applying GRPO.

### 3.1 FROM A SINGLE LOGIT UPDATE TO THE ENTROPY CHANGE

We consider a single decoding step where the policy $\pi$ produces a distribution over a vocabulary $\mathcal{V}$ of size $V$. Let $\mathbf{z} \in \mathbb{R}^V$ be the model's output logits. These logits are transformed into a probability distribution $\mathbf{p}$ via the softmax function, where $p_i = \frac{\exp(z_i)}{\sum_{j=1}^{V} \exp(z_j)} \forall i \in [V]$. The diversity of this probability distribution is measured by the token-level Shannon entropy (Shannon, 1948), which can be formally given as $H(\mathbf{p}) = -\sum_{i=1}^{V} p_i \log p_i$.

Throughout our analysis, we make the following standard assumptions for deriving first-order dynamics: (i) All probabilities $\{p_i\}$ are non-zero, as guaranteed by the softmax function; and (ii) Auxiliary regularization terms, including KL-divergence penalties, and explicit entropy bonuses, are considered inactive within RFT unless explicitly specified. (iii) We ignore tokens who trigger logit clipping as their gradients are set to zero and contribute no change to entropy.

The analysis begins with a fundamental operation in model update, i.e., updating the logit of a single token. We model this as a perturbation as $\delta \mathbf{z} = \varepsilon \cdot \mathbf{e}_k$, where $\mathbf{e}_k$ is the standard basis vector for the $k$-th token, and $\varepsilon$ is the change caused by the optimization process. The sign of $\varepsilon$, i.e., $\text{sign}(\varepsilon)$, represents the direction of the update: $\text{sign}(\varepsilon) = +1$ corresponds to rewarding the token (increasing the logits), while $\text{sign}(\varepsilon) = -1$ corresponds to penalizing it (decreasing the logits). The following lemma quantifies how this logit perturbation propagates to the probability distribution.

**Lemma 1.** *Given a logit perturbation $\delta \mathbf{z} = \varepsilon \cdot \mathbf{e}_k$ on $k$-th token $a^k$ in the vocabulary, the resulting first-order change in the probability distribution $\mathbf{p}$ is given by:*

$$\delta p_k = \varepsilon p_k(1 - p_k) \quad and \quad \delta p_i = -\varepsilon p_i p_k, \forall i \in [V], i \neq k. \tag{2}$$

*Proof.* The Jacobian of the softmax function is $\frac{\partial p_i}{\partial z_j} = p_i \left( \mathbf{1}\{i = j\} - p_j \right)$, where $\mathbf{1}\{\cdot\}$ is the indicator function. The first-order change $\delta p_i$ is given by the Taylor expansion $\delta p_i = \sum_{j=1}^{V} \frac{\partial p_i}{\partial z_j} \delta z_j + O(\varepsilon^2)$. Since $\delta z_j = \varepsilon \cdot \mathbf{1}\{j = k\}, \forall j \in [V]$, we have $\delta p_i = \frac{\partial p_i}{\partial z_k} \varepsilon = \varepsilon \cdot p_i(\mathbf{1}\{i = k\} - p_k)$, which yields the results in equation 2. $\square$

An immediate consequence of Lemma 1 is that the relative change in probability is uniform for all unperturbed tokens. Based on equation 2, the relative changes in probabilities can be given by:

$$\frac{\delta p_k}{p_k} = \varepsilon(1 - p_k) \quad and \quad \frac{\delta p_i}{p_i} = -\varepsilon p_k, \quad \forall i \in [V], i \neq k. \tag{3}$$

The aforementioned analysis shows that, when the probability of token $a^k$ is adjusted, its probability mass is redistributed proportionally from (or to) all other tokens. This aligns with the observation in previous works (Ren & Sutherland, 2025).

Building upon this insight, we can now derive a closed-form expression for the first-order change in entropy. We first define a key quantity that would determine the direction of this change. Let the entropy change discriminator for token on position $t$ be defined as $S_i^t \triangleq p_i^t(H^t + \log p_i^t)$, where the subscript $t$ is omitted when not causing confusion. In particular, assuming token $a^k$ is chosen at this position, the corresponding discriminator is denoted as $S_*^t \triangleq S_k^t$.

**Theorem 1.** *The first-order change in entropy, denoted by $\Delta H$, under the perturbation $\delta \mathbf{z} = \varepsilon \mathbf{e}_k$ is given by:*

$$\Delta H = -\varepsilon S_* + O(\varepsilon^2). \tag{4}$$

*Proof.* The first-order Taylor expansion of entropy $H$ around $\mathbf{p}$ can be given as:

$$\Delta H = H(\mathbf{p} + \delta \mathbf{p}) - H(\mathbf{p}) = \sum_{i=1}^{V} \frac{\partial H}{\partial p_i} \delta p_i + O(\|\delta \mathbf{p}\|^2). \tag{5}$$

Since $\frac{\partial H}{\partial p_i} = -(1 + \log p_i)$ and conservation of probability implies $\sum_i \delta p_i = 0$, we have:

$$\Delta H = -\sum_{i=1}^{V} (1 + \log p_i)\delta p_i + O(\varepsilon^2) = -\sum_{i=1}^{V} \log p_i \delta p_i + O(\varepsilon^2).$$

Substituting the expressions for $\delta p_i$ from Lemma 1, $\Delta H$ can be simplified as:

$$\Delta H = -\varepsilon \Big( p_k(1 - p_k) \log p_k - p_k \sum_{i \neq k} p_i \log p_i \Big) + O(\varepsilon^2) = -\varepsilon \, p_k \big( H + \log p_k \big) + O(\varepsilon^2),$$

which completes the proof. $\qquad \square$

**Implications** Theorem 1 provides a simple yet effective criterion for determining how a single-token update affects policy entropy. The direction of entropy change is dictated by the sign of two factors: the update direction $\varepsilon$ and the discriminator $S_*$. The sign of the discriminator $S_*$ depends on the relationship between the token's probability $p_k$ and the overall entropy $H(\mathbf{p})$:

$$\operatorname{sign}(S_*) = \operatorname{sign}\left(H(\mathbf{p}) + \log p_k\right) = \operatorname{sign}\left(p_k - e^{-H(\mathbf{p})}\right). \tag{6}$$

Consequently, rewarding a token ($\operatorname{sign}(\varepsilon) = +1$) increases entropy if its probability $p_k < e^{-H(\mathbf{p})}$ (a relatively low-probability token) and decreases entropy if $p_k > e^{-H(\mathbf{p})}$ (a relatively high-probability token). The relationship is reversed when a token is penalized ($\operatorname{sign}(\varepsilon) = -1$).

This microscopic analysis is the foundational building block for understanding entropy dynamics in RFT. Given that most existing RFT algorithms (Shao et al., 2024; Yu et al., 2025; Zheng et al., 2025) apply an update signal of the same direction to all tokens within a single response, our analysis explains the common empirical observation of rapid entropy collapse when models are consistently rewarded for generating high-probability and "safe" responses (He et al., 2025), which can lead to a gradual loss of the model's exploratory capabilities.

### 3.2 EXTENSION TO A GRPO OPTIMIZATION STEP

Beyond the above single-logit analysis, we extend our framework to model the entropy change resulting from a GRPO optimization step introduced in Section 2. Recall the training objective function of GRPO in equation 1, for a chosen token $a^k$ with token id $k$, its contribution to the whole training target can be given as:

$$\frac{\mathbf{p}_k}{p_k'} \cdot A \,,$$

where $\mathbf{p}_k$ denotes the current model distribution in the sampled position and $p_k'$ is the probability of the sampled token under the sampling model distribution. Therefore, its contribution to the GRPO training loss can be given by a surrogate loss:

$$\mathcal{L}(\mathbf{z}) = r \cdot A \cdot \log p_k(\mathbf{z}), \tag{7}$$

where $A$ represents the (group-relative) advantage and $r = \pi_\theta(a^k)/\pi_{\theta_{\text{sample}}}(a^k)$ is the importance sampling ratio.

A single gradient update step with learning rate $\eta$ results in a first-order change to the logits $\mathbf{z}$:

$$\delta \mathbf{z} = \eta \, \nabla_{\mathbf{z}} L = \alpha \, \nabla_{\mathbf{z}} \log p_k \,, \tag{8}$$

where we define $\alpha = \eta r A$ as the effective step size.

Recall the Jacobian of the softmax function in Lemma 1, $\nabla_{\mathbf{z}} p_k = p_k(\mathbf{e}_k - \mathbf{p})$. Therefore, we have:

$$\delta \mathbf{z} = \alpha \, \nabla_{\mathbf{z}} \log p_k = \alpha \frac{1}{p_k} \nabla_{\mathbf{z}} p_k = \alpha(\mathbf{e}_k - \mathbf{p}). \tag{9}$$

**Theorem 2.** *Let $S_i$ be the entropy discriminant for token $i$, and let its expectation over the policy distribution be $\mathbb{E}_{i\sim\mathbf{p}}[S_i] = \sum_{i=1}^{V} p_i S_i$. The first-order change in entropy of a token $H(\mathbf{p})$ satisfies:*

$$\Delta H = -\alpha \left(S_* - \mathbb{E}_{i\sim\mathbf{p}}[S_i]\right) + O(\alpha^2). \tag{10}$$

*Proof.* Recall the token-wise objective defined in equation 7 and a single update step defined in equation 8. Since $\mathbf{p} = \mathrm{softmax}(\mathbf{z})$, its Jacobian matrix is $J = \frac{\partial \mathbf{p}}{\partial \mathbf{z}} = \mathrm{diag}(\mathbf{p}) - \mathbf{p}\mathbf{p}^\top$, yielding the following equations:

$$\delta\mathbf{p} = J\,\delta\mathbf{z} = \left(\mathrm{diag}(\mathbf{p}) - \mathbf{p}\mathbf{p}^\top\right)\alpha(\mathbf{e}_k - \mathbf{p}) = \alpha\left[\mathbf{p}\odot(\mathbf{e}_k - \mathbf{p}) - \mathbf{p}\left(p_k - \|\mathbf{p}\|_2^2\right)\right],$$

$$\delta p_i = \alpha\left[p_i(\mathbf{1}\{i = k\} - p_i) - p_i(p_k + \|\mathbf{p}\|_2^2)\right].$$

As the first-order entropy change is given in equation 5, we substitute $\delta p_i$ and apply $\sum_i p_i \log p_i = -H$ to equation 5, which yields:

$$\Delta H = \alpha\left[\sum_i p_i^2\left(H + \log p_i\right) - p_k\left(H + \log p_k\right)\right] + O(\alpha^2) = -\alpha\left[S_* - \mathbb{E}_{i\sim\mathbf{p}}[S_i]\right] + O(\alpha^2).$$

The proof is completed by applying the definition of $S_i$ and $\mathbb{E}_{i\sim\mathbf{p}}[S_i]$. $\qquad\square$

**Implications** Theorem 2 reveals a crucial distinction from the single-logit case. With a GRPO optimization step, the entropy change is no longer governed by the absolute value of the entropy discriminant score $S_*$, but by its deviation from the policy-weighted expectation $\mathbb{E}_{i\sim\mathbf{p}}[S_i]$, which acts as a dynamic baseline. Entropy decreases if we reward (positive $\alpha$) a token $a^k$ whose score $S_*$ is above the baseline, and it increases when $S_*$ is below the baseline. The relationship can be reversed when we penalize (negative $\alpha$) a token.

Moving a step forward, we provide two corollaries derived from Theorem 2.

**Corollary 1.** *To a first-order approximation, with on-policy sampling, the expected entropy change factor $S_k - \mathbb{E}_{i\sim p}[S_i]$ of a token within GRPO optimization is zero, i.e.,*

$$\mathbb{E}_{k\sim\mathbf{p}}\left[S_* - \mathbb{E}_{i\sim\mathbf{p}}[S_i]\right] = 0. \tag{11}$$

**Corollary 2.** *For on-policy GRPO training with a batch, the expected value of entropy change factor $S_*^t - \mathbb{E}_{i\sim\mathbf{p}_t}[S_i^t]$ over the batch of tokens $\mathcal{T}_\mathcal{B}$ is zero:*

$$\mathbb{E}_{t\in\mathcal{T}_\mathcal{B}}\left[S_*^t - \mathbb{E}_{i\sim\mathbf{p}_t}[S_i^t]\right] = 0. \tag{12}$$

We provide the proofs for these two corollaries in Appendix A. These corollaries demonstrate that, from both the vocabulary and batch perspectives, the discriminant score $S_*$ possesses a favorable decentralization property under on-policy sampling. Therefore, imposing constraints on tokens based on the value of $S_*$ relative to its expectation offers a simple and direct approach to regulating entropy dynamics. Based on such analysis, we propose two methods for constraining entropy in Section 4.1.

Considering the advantage term, which depends on the token distribution, we can extend Corollaries 1 and 2 to the expectation of the full $\Delta H$ expression.

**Corollary 3.** *For on-policy GRPO training, the first-order expectation of the token-wise entropy change is given by:*

$$\mathbb{E}_{k\sim\mathbf{p}}[\Delta H] = -\eta\,\mathrm{Cov}_{k\sim\mathbf{p}}(A, S_* - \mathbb{E}_{i\sim\mathbf{p}}[S_i]). \tag{13}$$

**Corollary 4.** *For on-policy GRPO training with a batch, the first-order batch-wise entropy change of tokens $\mathcal{T}_\mathcal{B}$ is given by:*

$$\Delta H_{\mathcal{T}_\mathcal{B}} = -\eta\,\mathrm{Cov}_\mathcal{B}(A, S_* - \mathbb{E}_{i\sim\mathbf{p}}[S_i]). \tag{14}$$

Detailed proofs, experimental observations, and discussions are provided in Appendix C.

# 4 Bridging Entropy Dynamics to Entropy Control Methods

## 4.1 Entropy Discriminator Guided Clipping

The theoretical analysis provides novel insights into the relationships between the discriminator score $S_*^t$ and the entropy dynamics in RFT. Upon this, we can effectively identify tokens within a training batch that exert a disproportionate impact on entropy changes, enabling us to selectively mitigate the influence of such outlier tokens for achieving fine-grained and flexible control over the entropy regularization throughout the training process.

Inspired by Theorem 1, we propose a simple yet effective batch-level clipping method.

**Algorithm 1.** (*Clip$_\mathcal{B}$: Batch-Normalized Entropy-Discriminator Clipping*): Let $\mathcal{T}_\mathcal{B}$ denote the set of all tokens in the responses of a given batch $\mathcal{B}$. We first compute the batch-wise mean of the discriminator scores, $\bar{S} = \mathbb{E}_{t \in \mathcal{T}_\mathcal{B}}[S_*^t]$, and the corresponding standard deviation, $\sigma = \sqrt{\mathrm{Var}_{t \in \mathcal{T}_\mathcal{B}}[S_*^t]}$. During the RFT process, we only preserve the gradients associated with those tokens that satisfy a specific condition by applying the following mask $m_t$ to each token $t$:

$$m_t = \mathbf{1}\left\{ -\mu^- \sigma \leq S_*^t - \bar{S} \leq \mu^+ \sigma \right\}. \tag{15}$$

Here $\mu^+$ and $\mu^-$ are used to control the clipping threshold based on the degree of outlierness. This algorithm identifies the effects of each token on entropy change and filters out tokens that contribute to severe fluctuations in $\Delta H$. This is accomplished by simply examining the logits of response tokens, combined with the proposed batch-level normalization. This operation requires minimal computation on scalar values rather than high-dimensional tensors, thus introducing negligible additional computational cost and allowing for easy integration into existing training frameworks.

Moreover, Theorem 2 provides a more precise characterization of the entropy change, particularly in the context of using practical GRPO. This analysis motivates us to derive the vocabulary-normalized entropy-discriminator clipping method.

**Algorithm 2.** (*Clip$_\mathcal{V}$: Vocabulary-Normalized Entropy-Discriminator Clipping*):

For each token $t$ in a batch $\mathcal{T}_\mathcal{B}$, we first define its vocabulary-centered score as $S_c^t = S_*^t - \mathbb{E}_{i \sim \mathbf{p}_t}[S_t^i]$, where $\mathbf{p}_t$ is the policy's predictive distribution over the vocabulary $\mathcal{V}$ at step $t$. We then compute the standard deviation of these centered scores across the batch: $\sigma' = \sqrt{\mathrm{Var}_{t \in \mathcal{T}_\mathcal{B}}[S_c^t]}$. As established in Corollary 2, the batch-mean of these centered scores, $\mathbb{E}_{t \in \mathcal{T}_\mathcal{B}}[S_c^t]$, approximates zero. This simplifies the clipping condition. The mask for a token $t$ is thus defined as:

$$m_t = \mathbf{1}\left\{ -\mu^- \sigma' \leq S_*^t - \mathbb{E}_{i \sim \mathbf{p}_t}[S_i^t] \leq \mu^+ \sigma' \right\}. \tag{16}$$

In Clip$_\mathcal{V}$, the computation of $\mathbb{E}_{i \sim \mathbf{p}_t}[S_i^t]$ introduces some computational overhead. Fortunately, the quantities required to compute this term, such as the policy's logits over the full vocabulary, are often available as intermediate results from the forward pass used for entropy and log-probability calculations. This allows us to evaluate the expectation at a relatively low additional cost.

In Section 5.3, we provide empirical studies on the effectiveness of Clip$_\mathcal{B}$ and Clip$_\mathcal{V}$.

## 4.2 Interpreting Existing Methods through Entropy Dynamics

Recent works have proposed various entropy-based methods to enhance training stability and effectiveness. These methods, however, are often developed from heuristic principles and can necessitate labor-intensive hyperparameter tuning without clear theoretical guidance. To provide a better understanding of their underlying mechanisms, we re-examine these methods through the lens of our entropy dynamics analysis (refer to Section 3).

We categorize the related studies into three groups: (i) *Clipping Mechanisms*, which stabilizes the optimization by constraining the updates of token probability. Representative works include the clipping operation in GRPO (Guo et al., 2025) and the clip-higher method in DAPO (Yu et al., 2025). (ii) *Entropy Regularization*, which regularizes updates to tokens with high entropy, as proposed by Wang et al. (2025). (iii) *Probability Weighted Updating*, which constrains the updates based on token probabilities, exemplified by methods from (He et al., 2025; Yang et al., 2025b).

Before we conduct the investigation and interpretation, let's first recall Theorem 2 and examine, from a statistical perspective, the relationship it reveals between two factors often considered by the methods above, i.e., probability and entropy. The first term in Theorem 2, $S_* = p_k(H + \log p_k)$, directly associates these two factors. For tokens sampled with high probability, $S_*$ tends to be larger; similarly, tokens sampled in positions with high entropy also have larger $S_*$. Tokens sampled with larger $S_*$ are more likely to obtain a positive value when calculating the deviation from the expectation $\mathbb{E}_{i \sim k}[S_i]$, as the expectation represents an average value under the current model's sampling distribution.

As a result, when considering changes in entropy $\Delta H$, for positive samples, higher probability and lower token entropy are often associated with a decrease in entropy. In contrast, lower probability and higher entropy are often linked to an increase in entropy. For negative samples, the trend is reversed.

**Clipping Mechanisms** The clipping mechanism in GRPO can be formulated as a gradient mask for the $t$-th token in response $i$:

$$M_{i,t} = \mathbf{1}\{A_i > 0, r_{i,t}(\theta) \leq 1 + \epsilon_{\text{high}}\} + \mathbf{1}\{A_i < 0, r_{i,t}(\theta) \geq 1 - \epsilon_{\text{low}}\}, \quad (17)$$

where $r_{i,t}(\theta) = \frac{\pi_\theta(o_{i,t}|q,o_{i,<t})}{\pi_{\text{sample}}(o_{i,t}|q,o_{i,<t})}$ is the importance ratio. The clipping mechanism prevents an excessive increase in probability for tokens in positive samples and an excessive decrease for tokens in negative samples. Due to the nature of the importance ratio, this mechanism predominantly affects tokens with low initial probabilities under the sampling policy.

Empirical statistics from (Yu et al., 2025) show that clipped tokens typically have a maximum probability of around 0.15. Across a training trajectory where overall token entropy declines. As we analyzed above, these low-probability tokens are associated with the condition $S_* - \mathbb{E}[S_i] < 0$. For these tokens, the sign of the entropy change is given by $\text{sign}(\Delta H) = -\text{sign}(\varepsilon)\text{sign}(S_* - \mathbb{E}_{i \sim \mathbf{p}}[S_i]) = \text{sign}(\varepsilon)$. Consequently, updates on positive samples $(\text{sign}(\varepsilon) = +1)$ tend to increase token entropy, while updates on negative samples $(\text{sign}(\varepsilon) = -1)$ tend to decrease it. The overall entropy dynamics is a superposition of these two effects: in most cases, it manifests as a rapid decline in entropy, while in others, it exhibits relatively complex fluctuations (Liu et al., 2025).

The clip-higher method in DAPO, which sets a larger $\epsilon_{\text{high}}$ for positive samples, is correspond to this insight. By relaxing the clipping constraint for positive samples, it preserves their entropy-increasing updates. This targeted intervention counteracts the natural tendency of entropy to decline during RFT, thereby promoting exploration and improving model performance. These observations align with our theoretical framework.

**Entropy Regularization** Entropy regularization refers to methods that compute gradients only for a certain proportion of tokens with high entropy. For example, Wang et al. (2025) demonstrates improved performance by applying updates to only the top 20% of tokens with the highest entropy.

As we analyzed above, high token entropy corresponds to a condition where our theoretical quantity $S_* - \mathbb{E}_{i \sim \mathbf{p}}[S_i]$ is likely to be positive. According to Theorem 2, for these tokens, the updates on positive samples would decrease entropy, while updates on negative samples would increase it. The net effect on entropy is therefore determined by the balance between these two opposing forces. The empirical results in (Wang et al., 2025), which show that as the proportion of selected high-entropy tokens is varied, the overall entropy first increases and then decreases relative to a baseline, provide strong evidence for this trade-off.

**Probaility Weighted Updating** Similar to entropy regularization, Probaility Weighted Updating methods constrain or scale token updates based on their probabilities. For example, He et al. (2025) proposes to assign higher weights to positive samples with low probability. In the context of our analysis, low-probability tokens are associated with $S_* - \mathbb{E}_{i \sim \mathbf{p}}[S_i] < 0$. When these tokens are part of a positive sample, the expected change in entropy is positive. By amplifying the updates for this specific subset of tokens, the method explicitly promotes gradients that increase token entropy, alleviating the entropy collapse issue. The provided experimental results support this conclusion.

In summary, our analysis offers a unified view for understanding the underlying mechanics of existing methods, which function by amplifying the effects of tokens contributing to entropy increase or suppressing those leading to entropy decrease, thereby preventing entropy collapse in RFT.

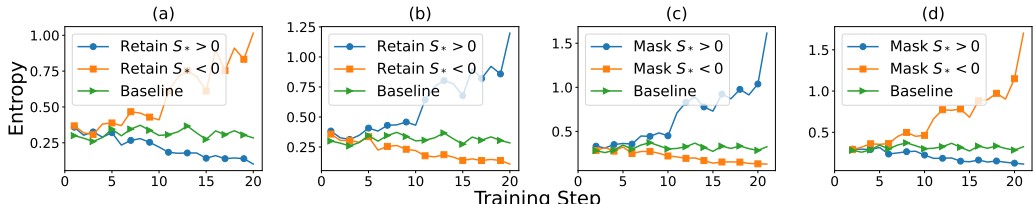

Figure 1: We retain or mask the gradients of tokens satisfying $S_* > 0$ or $S_* < 0$, respectively. The resulting entropy changes are shown in (a,c) for positive samples, and (b,d) for negative samples.

## 5 EXPERIMENTS

In this section, we conduct experiments to provide empirical evidence for the theoretical analysis in Section 3, and to demonstrate the effectiveness of the proposed methods in Section 4.1.

### 5.1 SETTINGS

We select the Qwen2.5-7B-Instruct (Yang et al., 2024) and Qwen3-4B-Base (Yang et al., 2025a) as our base models for RFT, utilizing the DAPO-Math-17k dataset (Yu et al., 2025) as our training set. Following previous studies (Lightman et al., 2023), we exclude 500 questions from the training set to form the validation set (denoted by DAPO500). We filter out samples from the training set with excessively high ($\geq 15/16$) or low ($\leq 1/16$) pass rates, as evaluated by Qwen2.5-7B-Instruct.

For evaluation, we adopt two challenging mathematical datasets, i.e., AIME24 and AIME25, to form our test set. We adopt the Avg@32/Pass@32 evaluation metrics for AIME24 and AIME25, and Avg@8/Pass@8 for DAPO500. Here Avg@K denotes the average accuracy across K responses for each question, while Pass@K represents the probability that at least one of K responses is correct.

### 5.2 EMPIRICAL OBSERVATIONS OF THE ENTROPY DYNAMICS

We first provide empirical evidence supporting Theorem 1, which posits a close relationship between the discriminator score $S_*$ and the direction of change in token entropy, i.e., $\text{sign}(\Delta H)$. Specifically, during the training process, we selectively update the loss associated with tokens exhibiting $S_* > 0$ or $S_* < 0$. The standard training process serves as our baseline for comparison. For clear observations, we apply these selective updates to positive (rewarding) and negative (punishing) samples separately, presenting the results in Figures 1(a) and 1(b), respectively. These experimental results align with our analysis. For example, in Figure 1(a), when we only retain updates for tokens with $S_* > 0$ in positive samples, we observe a decrease in entropy consistent with $\text{sign}(\Delta H) = -\text{sign}(\varepsilon) \cdot \text{sign}(S_*) < 0$. Conversely, retaining updates for tokens with $S_* < 0$ induces an increment in entropy. Such a phenomenon is precisely reversed in Figure 1(b) as we apply these operations to negative samples.

To further probe this relationship, we investigate a practical scenario where we mask the gradients of tokens that satisfy specific conditions during the training process. Similarly, as shown in Figure 1(c), when the gradients associated with tokens in positive samples that satisfy $S_* > 0$ are masked (these are believed to contribute to entropy decrease), the entropy increases uncontrollably. Conversely, masking the gradients of tokens that satisfy $S_* < 0$ leads to a continuous decrease in entropy. Figure 1(d) illustrates that performing the same masking operations on negative samples results in the opposite behavior. These experimental results further confirm our analysis, which suggests that the sign of the discriminator score $S_*$ is a reliable predictor of tokens' influence on entropy dynamics within RFT.

In Figure 3, we illustrate the distribution of $S_*$ within a training batch and its deviation from its sampling expectation, as involved in Theorem 2. The value of $\mathbb{E}_{t \sim \mathcal{T}_\mathcal{B}}[S_*^t - \mathbb{E}_{i \sim \mathbf{p}}[S_i^t]]$ is three orders of magnitude smaller than that of $\mathbb{E}_{t \sim \mathcal{T}_\mathcal{B}}[S_*^t]$, and approaches zero, effectively validating Corollary 2.

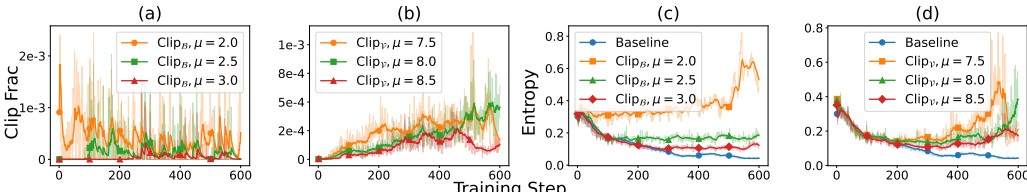

Figure 2: The effects of $\text{Clip}_{\mathcal{B}}$ and $\text{Clip}_{\mathcal{V}}$ with different $\mu$ in controlling clip fraction and entropy.

Table 1: Comparisons between vanilla GRPO and the proposed methods on Avg@K and Pass@K.

| Method | AIME24 | | AIME25 | | DAPO500 | |
|---|---|---|---|---|---|---|
| | Avg@K | Pass@K | Avg@K | Pass@K | Avg@K | Pass@K |
| Qwen2.5-7B-Inst | 11.35 | 36.67 | 6.67 | 33.33 | 31.55 | 70.2 |
| GRPO | 16.88 | 50.00 | 15.42 | 50.00 | 48.03 | 76.8 |
| GRPO+$\text{Clip}_{\mathcal{B}}$ | **19.69** (+2.81) | **56.67** (+6.67) | **16.35** (+0.93) | 53.33 (+3.33) | **49.68** (+1.65) | **80.2** (+3.4) |
| GRPO+$\text{Clip}_{\mathcal{V}}$ | 18.12 (+1.24) | 53.33 (+3.33) | 15.94 (+0.52) | **56.67** (+6.67) | 49.65 (+1.62) | 79.0 (+2.2) |
| Qwen3-4B-Base | 9.06 | 40.00 | 7.71 | 33.33 | 27.82 | 69.0 |
| GRPO | 20.73 | 53.33 | 18.96 | 56.67 | 50.38 | 81.2 |
| GRPO+$\text{Clip}_{\mathcal{B}}$ | 20.83 (+0.10) | 53.33 (+0.00) | 19.48 (+0.52) | 53.33 (-3.34) | 49.70 (-0.68) | **82.4** (+1.2) |
| GRPO+$\text{Clip}_{\mathcal{V}}$ | **21.56** (+0.83) | **56.67** (+3.34) | **20.42** (+1.46) | **60.00** (+3.33) | **50.95** (+0.57) | 82.0 (+0.8) |

## 5.3 Effects of Entropy Discriminator Clipping Methods

In this subsection, we validate the effectiveness of the clipping methods proposed in Section 4.1, including $\text{Clip}_{\mathcal{B}}$ and $\text{Clip}_{\mathcal{V}}$. Considering that entropy empirically exhibits a clear decreasing trend within RFT, we choose negative samples as the primary focus and apply our clipping methods to mask the losses of specific tokens. As shown in Figures 2(a) and 2(b), the hyper-parameter $\mu$ in the $\text{Clip}_{\mathcal{B}}$ and $\text{Clip}_{\mathcal{V}}$ methods provides effective control over the number of clipped tokens (a larger $\mu$ indicates a smaller clip proportion), thereby supporting flexible adjustment of the intervention intensity. In Figures 2(c) and 2(d), we illustrate the effects of the clipping methods in controlling entropy with different values of $\mu$. We observe that both $\text{Clip}_{\mathcal{B}}$ and $\text{Clip}_{\mathcal{V}}$ successfully mitigate the entropy decay to excessively low levels, as seen in the baseline (the standard RFT training).

Existing RFT studies (Yu et al., 2025; Liao et al., 2025) suggest that maintaining a certain level of entropy can retain the model's exploration capabilities, leading to better model performance. Therefore, we validate the performance of models trained using $\text{Clip}_{\mathcal{B}}$ and $\text{Clip}_{\mathcal{V}}$, as summarized in Table 1. These results demonstrate that both $\text{Clip}_{\mathcal{B}}$ and $\text{Clip}_{\mathcal{V}}$ achieve outperformance compared to standard GRPO in most scenarios, confirming their effect in preserving model exploration by controlling entropy. The performance gains are particularly noteworthy on the more difficult AIME24 and AIME25 datasets, suggesting that by preventing the model from prematurely converging to a low-entropy state, our methods enable the model to find correct solutions for complex problems.

## 5.4 Analysis of Exploration versus Exploitation

We compare the model performance on the Pass@K metric against the Avg@K metric. While Avg@K measures the average correctness across all rollouts, Pass@K measures the ability to find at least one correct solution within $K$ attempts. A larger gain in Pass@K would indicate that the model is encouraged to generate diverse responses for solving problems (exploration), rather than solely following similar rewarded solutions (which would significantly boost Avg@K, i.e., exploitation). As shown in Table 1, our methods achieve substantially more significant improvements in Pass@K compared to Avg@K across all datasets. These results confirm that stabilizing entropy with the proposed clipping method fosters greater solution diversity and encourages the model to discover correct reasoning paths for a wider array of problems.

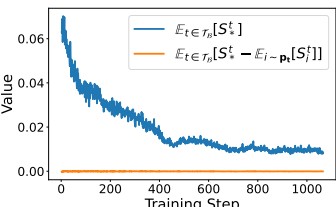

Figure 3: The batch-averaged value of $S_*$ and $S_* - \mathbb{E}_{i \sim \mathbf{p}}[S_i]$.

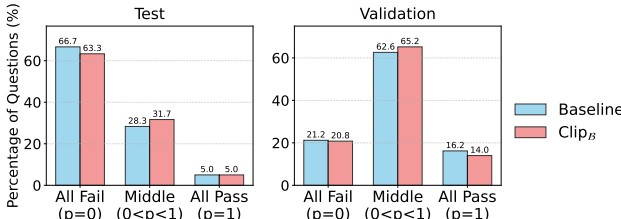

Figure 4: Comparison between $\text{Clip}_\mathcal{B}$ and vanilla GRPO on the distribution of problem pass rates.

Moreover, we further conduct a study on the distribution of pass rates among multiple rollouts for individual problems. Taking the Qwen-2.5-7B-Instruct model and the $\text{Clip}_\mathcal{B}$ method as an example, we illustrate the results in Figure 4. For the standard GRPO, the proportion of problems that are completely solved or completely failed is significantly higher than that of $\text{Clip}_\mathcal{B}$. This indicates that GRPO excessively prioritizes exploitation while neglecting the importance of exploration. Conversely, $\text{Clip}_\mathcal{B}$ focuses more on encouraging exploration, resulting in a pass rate distribution that is more concentrated around the middle range. This suggests that the performance gains achieved by our method stem from encouraging the model to explore solutions for a broader range of problems, rather than simply memorizing easier problems that could be solved with higher certainty.

## 6 RELATED WORKS

Reinforcement fine-tuning (RFT) has been widely adopted in tuning LLMs, including representative methods such as GRPO (Guo et al., 2025), DAPO (Yu et al., 2025), and GSPO (Zheng et al., 2025). To enhance LLM performance, many strategies have been proposed from diverse aspects (Liu et al., 2025; Hu, 2025; Yu et al., 2025). Among these works, some tricks are developed based on the influence of entropy on model behavior, such as explicitly entropy regularization (Hu et al., 2025), entropy-based token selection (Wang et al., 2025), flexible clipping schemes (Yu et al., 2025), and others (Liao et al., 2025). To better understand the role of entropy, a recent work (Cui et al., 2025) theoretically analyzed entropy changes, revealing the mechanisms linking entropy to the distribution of model sampling. In addition, they formulated a model characterizing the relationship between model performance and entropy variation, highlighting the importance of studying entropy dynamics in the context of reinforcement fine-tuning.

## 7 CONCLUSIONS

In this study, we focus on a theoretical framework to provide a principled understanding of entropy dynamics in RFT. We quantify the entropy change and further extend this analysis to a practical GRPO optimization step, revealing that entropy fluctuations arise from the combined effect of a token's update direction, its probability, and the policy entropy. These insights offer explanations for the commonly observed entropy collapse phenomenon, guide the development of entropy controlling strategies, and unify the interpretation of existing entropy-based methods. We hope that the theoretical framework can foster a clear understanding of the underlying mechanisms of entropy dynamics in RFT, thereby accelerating progress in the field.

## REPRODUCIBILITY STATEMENT

We provide the experimental setup and details in Section 5 and Appendix B. All the datasets and models used in this work can be publicly accessed. The source code will be open-sourced after the public review stage.

## ETHICS STATEMENT

This work aims to investigate the large language models, with potential benefits in improving model performance. We do not collect or release datasets, and our experiments are based on publicly available models. We believe that advancing model efficiency contributes positively to sustainable and accessible artificial intelligence.

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

# A    PROOF OF COROLLARIES

**Corollary 1.** *To a first-order approximation, with on-policy sampling, the expected entropy change of a token within GRPO optimization is zero, i.e.*

$$\mathbb{E}_{k\sim\mathbf{p}}\left[S_* - \mathbb{E}_{i\sim\mathbf{p}}[S^i]\right] = 0.$$

*Proof.* We derive the results as follows:

$$\mathbb{E}_{k\sim\mathbf{p}}\left[S_* - \mathbb{E}_{i\sim\mathbf{p}}[S^i]\right] = \mathbb{E}_{k\sim\mathbf{p}}\left[p_k\left(H + \log p_k\right) - \sum_{i=1}^{V} p_i^2\left(H + \log p_i\right)\right]$$

$$= \sum_{k=1}^{V} p_k^2\left(H + \log p_k\right) - \sum_{i=1}^{V} p_i^2\left(H + \log p_i\right)$$

$$= 0.$$

$\square$

**Corollary 2.** *For on-policy GRPO training with a batch, the expected value of entropy change factor $S_*^t - \mathbb{E}_{i\sim\mathbf{p}_t}[S_i^t]$ over the batch of tokens $\mathcal{T}_{\mathcal{B}}$ is zero:*

$$\mathbb{E}_{t\in\mathcal{T}_{\mathcal{B}}}\left[S_*^t - \mathbb{E}_{i\sim\mathbf{p}_t}[S_i^t]\right] = 0. \tag{18}$$

*Proof.* For each token $t \in \mathcal{T}_{\mathcal{B}}$, let $\mathbf{p}_t = (p_t^1, \ldots, p_t^V)$ be the on-policy token distribution, $H_t = -\sum_{i=1}^{V} p_t^i \log p_t^i$, and $S_t^i := p_t^i\left(H_t + \log p_t^i\right)$. Draw the action index on-policy: $K_t \sim \text{Cat}(\mathbf{p}_t)$. Then, conditioning on $\mathbf{p}_t$,

$$\mathbb{E}\left[S_t^{K_t}\,\Big|\,\mathbf{p}_t\right] = \sum_{i=1}^{V} p_t^i\,S_t^i = \sum_{i=1}^{V}(p_t^i)^2\left(H_t + \log p_t^i\right) = \mathbb{E}_{i\sim\mathbf{p}_t}[S_t^i].$$

Hence $\mathbb{E}\left[S_t^{K_t} - \mathbb{E}_{i\sim\mathbf{p}_t}[S_t^i]\,\Big|\,\mathbf{p}_t\right] = 0$ for each token $t$. Averaging over the batch and using linearity of expectation,

$$\mathbb{E}\left[\frac{1}{|\mathcal{T}_{\mathcal{B}}|}\sum_{t\in\mathcal{T}_{\mathcal{B}}}\left(S_t^{K_t} - \mathbb{E}_{i\sim\mathbf{p}_t}[S_t^i]\right)\,\Bigg|\,\{\mathbf{p}_t\}_{t\in\mathcal{T}_{\mathcal{B}}}\right] = \frac{1}{|\mathcal{T}_{\mathcal{B}}|}\sum_{t\in\mathcal{T}_{\mathcal{B}}} 0 = 0.$$

Finally, applying the tower property removes the conditioning and yields the stated result.    $\square$

# B    DETAILED EXPERIMENT SETUP

All experiments are conducted on NVIDIA A100 and H20 GPUs. We implement the experiment with the Trinity-RFT (Pan et al., 2025) framework.

For the training process, we adopt the Adam optimizer with hyperparameters $(0.9, 0.999)$. We set the training batch size to 64, the number of rollouts to 16 for Qwen2.5-7B-Instruct and 8 for Qwen3-4B-Base, and employ a learning rate of $4 \times 10^{-7}$. The temperature is set to 1.0 for sampling rollouts and 0.7 for evaluation.

**Reward design**    The reward for each response is determined by its answer and format correctness, i.e., (1) $r_i = 1$ if the answer and format are correct; (2) $r_i = 0$ if the answer is correct but the format is not; (3) $r_i = -1.0$ if the answer and format are incorrect and the response is extremely long; and (4) $r_i = -0.1$ if the answer and format are incorrect but the length remain within the constraint.

# C    EXTENSION TO ADVANTAGE-AWARE ANALYSIS

In this section, we extend the findings of Corollary 1 and Corollary 2 to incorporate advantage estimation. We analyze the expectations of the full entropy change expression presented in Theorem 2 from two complementary perspectives: model sampling and batch averaging.

## C.1 EXTENSION OF COROLLARY 1 TO MODEL SAMPLING

We assume that at each position $t$, every token ID in the vocabulary has a latent advantage value, i.e., $A = \mathbf{A}(i)$ for $i \sim \mathbf{p}_t$. Building upon Theorem 2, we derive the following corollary regarding the expected entropy change.

**Corollary 3.** *For on-policy GRPO training, the first-order expectation of the token-wise entropy change is given by:*

$$\mathbb{E}_{k \sim \mathbf{p}}[\Delta H] = -\eta \operatorname{Cov}_{k \sim \mathbf{p}}(A, S_* - \mathbb{E}_{i \sim \mathbf{p}}[S_i]). \tag{19}$$

*Proof.* We begin by applying the result from Theorem 2. Recall that $\alpha = \eta r A$. Considering the constant $\eta$ and $r = 1$ in the on-policy setting, the first-order expectation of token entropy change under on-policy sampling can be given by:

$$\mathbb{E}_{k \sim \mathbf{p}}[\Delta H] = -\eta \mathbb{E}_{k \sim \mathbf{p}}\big[A(S_* - \mathbb{E}_{i \sim \mathbf{p}}[S_i])\big]. \tag{20}$$

We decompose the covariance in this equation, which gives:

$$\begin{aligned}
\mathbb{E}_{k \sim \mathbf{p}}[\Delta H] = -\eta \big\{ &\mathbb{E}_{k \sim \mathbf{p}}[A]\mathbb{E}_{k \sim \mathbf{p}}[S_*] + \operatorname{Cov}_{k \sim \mathbf{p}}(A, S_*) \\
&- \mathbb{E}_{k \sim \mathbf{p}}[A]\mathbb{E}_{k \sim \mathbf{p}}[\mathbb{E}_{i \sim \mathbf{p}}[S_i]] - \operatorname{Cov}_{k \sim \mathbf{p}}(A, \mathbb{E}_{i \sim \mathbf{p}}[S_i]) \big\}.
\end{aligned}$$

By Corollary 1, we have $\mathbb{E}_{k \sim \mathbf{p}}[S_*] - \mathbb{E}_{k \sim \mathbf{p}}[\mathbb{E}_{i \sim \mathbf{p}}[S_i]] = 0$. Substituting this into the equation above, we have:

$$\mathbb{E}_{k \sim \mathbf{p}}[\Delta H] = -\eta \operatorname{Cov}_{k \sim \mathbf{p}}(A, S_* - \mathbb{E}_{i \sim \mathbf{p}}[S_i]).$$

$\square$

**Implications.** Corollary 3, based on the on-policy policy gradient formula, provides a clean expression for entropy change. It decouples the advantage from the core entropy change term $S_* - \mathbb{E}_{i \sim \mathbf{p}}[S_i]$, and establishes their relationship through a covariance. In GRPO, however, the advantage for tokens not actually sampled is undefined and non-computable; therefore, Corollary 3 cannot be directly applied in algorithmic implementation. Nevertheless, it offers a theoretical potential for understanding entropy collapse in the GRPO training process.

In GRPO, the way advantages are obtained is coupled with the policy model distribution, which promotes entropy collapse. We will verify this hypothesis from a batch-level perspective in the next subsection.

## C.2 EXTENSION OF COROLLARY 2

The batch-level entropy is defined by the arithmetic average of token entropies within a batch:

$$H_{\mathcal{T}_{\mathcal{B}}} = \frac{1}{|\mathcal{T}_{\mathcal{B}}|} \sum_{t \in \mathcal{T}_{\mathcal{B}}} H_t$$

Therefore, batch-level entropy change is also the arithmetic average of token entropy change:

$$\Delta H_{\mathcal{T}_{\mathcal{B}}} = \frac{1}{|\mathcal{T}_{\mathcal{B}}|} \sum_{t \in \mathcal{T}_{\mathcal{B}}} \Delta H_t \tag{21}$$

Based on the above definition, we derive the following corollary:

**Corollary 4.** *For on-policy GRPO training with a batch, the first-order batch-wise entropy change of tokens $\mathcal{T}_{\mathcal{B}}$ is given by:*

$$\Delta H_{\mathcal{T}_{\mathcal{B}}} = -\eta \operatorname{Cov}_{\mathcal{B}}(A, S_* - \mathbb{E}_{i \sim \mathbf{p}}[S_i]). \tag{22}$$

*Proof.* Applying the results in Theorem 2 into equation 21 gives:

$$\Delta H_{\mathcal{T}_{\mathcal{B}}} = -\frac{1}{|\mathcal{T}_{\mathcal{B}}|} \sum_{t \in \mathcal{T}_{\mathcal{B}}} \alpha(S_* - \mathbb{E}_{i \sim \mathbf{p}_t}[S_i^t]) = -\mathbb{E}_{\mathcal{B}}[\alpha(S_* - \mathbb{E}_{i \sim \mathbf{p}}[S_i])], \tag{23}$$

where $\mathbb{E}_{\mathcal{B}}$ refers to statistical expectation (i.e., an arithmetic average over batch $\mathcal{B}$).

Recall the definition of $\alpha = \eta r A$: the learning rate $\eta$ is constant within a batch; $r$ is constantly 1 in the on-policy setting. The advantage $A$ estimated in the GRPO algorithm is NOT independent of the chosen token id in a batch $\mathcal{T}_\mathcal{B}$, i.e.,

$$\Delta H_{\mathcal{T}_\mathcal{B}} = -\eta \, \mathbb{E}_\mathcal{B}[A(S_* - \mathbb{E}_{i\sim\mathbf{p}}[S_i])].$$

We further apply covariance decomposition to $\Delta H_{\mathcal{T}_\mathcal{B}}$ within a training batch:

$$\Delta H_{\mathcal{T}_\mathcal{B}}/\eta = -\big\{\mathbb{E}_\mathcal{B}[A]\mathbb{E}_\mathcal{B}[S_*] + \mathrm{Cov}_\mathcal{B}(A, S_*) - \mathbb{E}_\mathcal{B}[A]\mathbb{E}_\mathcal{B}[\mathbb{E}_{i\sim\mathbf{p}}[S_i]] - \mathrm{Cov}_\mathcal{B}(A, \mathbb{E}_{i\sim\mathbf{p}}[S_i])\big\}.$$

According to Corollary 2, we have $\mathbb{E}_\mathcal{B}[S] - \mathbb{E}_\mathcal{B}[\mathbb{E}_{i\sim\mathbf{p}}[S_i]] = 0$, which gives:

$$\Delta H_{\mathcal{T}_\mathcal{B}}/\eta = -\mathrm{Cov}_\mathcal{B}(A, S_* - \mathbb{E}_{i\sim\mathbf{p}}[S_i]). \tag{24}$$

Finally, we multiply both sides of the above equation by $\eta$ to complete the proof. $\qquad\square$

**Implications.** Corollary 4 provides a computable form analogous to Corollary 3 from the batch perspective. We conduct a experiment to monitored the quantity $-\mathrm{Cov}_\mathcal{B}(A, S_* - \mathbb{E}_{i\sim\mathbf{p}}[S_i])$ during training. As shown in Figure 5, its value has a larger magnitude in the negative portion compared with the positive ones. This observation further validates the hypothesis in Appendix C.1. The model tends to obtain correct answers (i.e., $A > 0$) by producing "safe" responses, those with relatively high probability, for which $S_* - \mathbb{E}[S_i]$ tends to be positive, whereas exploratory behaviors are more likely to yield incorrect answers. This dynamic continually suppresses the model's propensity to explore diverse answers.

Algorithm 2 directly computes the factor $S_* - \mathbb{E}_{i\sim\mathbf{p}}[S_i]$, and masks those who contribute extremely significantly to the covariance expression.

For example, for negative samples where $A < 0$, Theorem 2 masks those tokens with large negative $S_* - \mathbb{E}_{i\sim\mathbf{p}}[S_i]$, who contributes a large negative factor to equation 14, stabling the change of entropy.

Algorithm 1 estimates this factor in a batch perspective, achieving better computational efficiency.

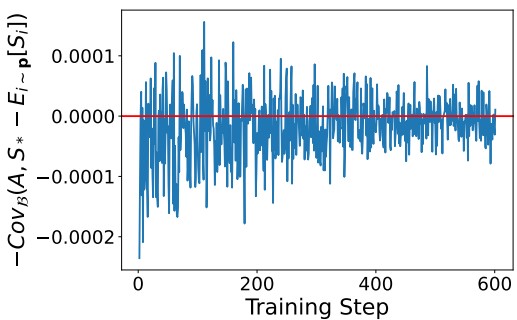

Figure 5: The value of $-\mathrm{Cov}_\mathcal{B}(A, S_* - \mathbb{E}_{i\sim\mathbf{p}}[S_i])$.

## D  EXTENSION TO OFF-POLICY SCENARIOS

The derivation of Theorem 2 is based on the general GRPO formulation and is not restricted to the on-policy setting. In this section, we extend Corollaries 1-4 to the off-policy scenario. When off-policy sampling is used, similar expressions can be obtained by utilizing the importance ratio $r = \pi_\theta/\pi_{\theta_\text{sample}}$.

**Corollary 1.1.** *To a first-order approximation, the expected entropy change factor $r(S_* - \mathbb{E}_{i\sim\mathbf{p}}[S_i])$ of a token within GRPO optimization is zero, i.e.,*

$$\mathbb{E}_{k\sim\mathbf{p}'}\left[r(S_* - \mathbb{E}_{i\sim\mathbf{p}}[S_i])\right] = 0,$$

where $\mathbf{p}'$ and $\mathbf{p}$ denote the sampling policy's and the current policy model's output distributions at token $t$, respectively.

*Proof.* We derive the results as follows:

$$\mathbb{E}_{k\sim\mathbf{p}'}\left[r(S_* - \mathbb{E}_{i\sim\mathbf{p}}[S_i])\right] = \mathbb{E}_{k\sim\mathbf{p}'}\left\{r\left[p_k(H + \log p_k) - \sum_{i=1}^{V} p_i^2(H + \log p_i)\right]\right\}$$

$$= \sum_{k=1}^{V} \frac{p_k}{p_k'} p_k' p_k(H + \log p_k) - \sum_{i=1}^{V} p_i^2(H + \log p_i)\sum_{k=1}^{V} \frac{p_k}{p_k'} p_k'$$

$$= (1 - \sum_{k=1}^{V} p_k)\sum_{i=1}^{V} p_i^2(H + \log p_i)$$

$$= 0.$$

□

**Corollary 2.1.** *For on-policy GRPO training with a batch, the expected value of entropy change factor $r(S_*^t - \mathbb{E}_{i\sim\mathbf{p}_t}[S_i^t])$ over the batch of tokens $\mathcal{T}_\mathcal{B}$ is zero:*

$$\mathbb{E}_{t\in\mathcal{T}_\mathcal{B}}[r(S_*^t - \mathbb{E}_{i\sim\mathbf{p}_t}[S_i^t])] = 0. \tag{25}$$

*Proof.* For each token in Batch $\mathcal{T}_\mathcal{B}$, define $\mathbf{p}_t$ as the distribution of the current policy, and $\mathbf{p}_t'$ as the distribution of the sampling policy. Considering one time step $t$, the selected token $K_t$ follows the distribution $\mathbf{p}_t'$, i.e., $K_t \sim \mathbf{p}_t'$. The importance ratio of token $K_t$ can be expressed as $r = \frac{\mathbf{p}_t(K_t)}{\mathbf{p}_t'(K_t)}$. The conditional expectation of each term under the sampling distribution $\mathbf{p}_t'$ is then given by:

$$\mathbb{E}_{K_t\sim\mathbf{p}_t'}\left[r\cdot(S_*^t - \mathbb{E}_{i\sim\mathbf{p}_t}[S_i^t]) \mid \mathbf{p}_t, \mathbf{p}_t'\right].$$

We expand this expression according to the definition of expectation:

$$\mathbb{E}_{K_t\sim\mathbf{p}_t'}\left[r\cdot(S_*^t - \mathbb{E}_{i\sim\mathbf{p}_t}[S_i^t]) \mid \mathbf{p}_t, \mathbf{p}_t'\right] = \sum_{k\in V} \mathbf{p}_t'(k)\cdot\frac{\mathbf{p}_t(k)}{\mathbf{p}_t'(k)}\cdot\left(S_k^t - \mathbb{E}_{i\sim\mathbf{p}_t}[S_i^t]\right)$$

$$= \sum_{k\in V} \mathbf{p}_t(k)\cdot\left(S_k^t - \mathbb{E}_{i\sim\mathbf{p}_t}[S_i^t]\right)$$

$$= \mathbb{E}_{k\sim\mathbf{p}_t}\left[S_k^t - \mathbb{E}_{i\sim\mathbf{p}_t}[S_i^t]\right].$$

According to Corollary 1, under the current policy distribution $\mathbf{p}_t$, the expectation of the difference between the discriminator score $S_*$ and its expectation is 0:

$$\mathbb{E}_{k\sim\mathbf{p}_t}[S_k^t] - \mathbb{E}_{k\sim\mathbf{p}_t}[\mathbb{E}_{i\sim\mathbf{p}_t}[S_i^t]] = \mathbb{E}_{i\sim\mathbf{p}_t}[S_i^t] - \mathbb{E}_{i\sim\mathbf{p}_t}[S_i^t] = 0.$$

Therefore, for any token $K_t$ in the Batch, the expected conditioned value of the entropy change factor is 0:

$$\mathbb{E}_{K_t\sim\mathbf{p}_t'}\left[r\cdot(S_*^t - \mathbb{E}_{i\sim\mathbf{p}_t}[S_i^t]) \mid \mathbf{p}_t, \mathbf{p}_t'\right] = 0 \tag{26}$$

Finally, taking the mean over Batch $\mathcal{T}_\mathcal{B}$ utilizing the linearity of expectation and tower property:

$$\mathbb{E}_{t\in\mathcal{T}_\mathcal{B}}[r(S_*^t - \mathbb{E}_{i\sim\mathbf{p}_t}[S_i^t])] = \mathbb{E}_{K_t\sim\mathbf{p}_t'}\left[\frac{1}{|\mathcal{T}_\mathcal{B}|}\sum_{t\in\mathcal{T}_\mathcal{B}} r(S_t^{K_t} - \mathbb{E}_{i\sim\mathbf{p}_t}[S_i^i])\Big|\{\mathbf{p}_t\}_{t\in\mathcal{T}_\mathcal{B}}\right] = \frac{1}{|\mathcal{T}_\mathcal{B}|}\sum_{t\in\mathcal{T}_\mathcal{B}} 0 = 0.$$

□

To extend the off-policy version of Corollaries 3 and 4, we leverage similar methods in proving Corollaries 1.1 2.1, i.e., replacing the results in Corollaries 1 and 2 with their off-policy counterparts. The following corollaries show the off-policy extensions of Corollaries 3 and 4.

**Corollary 3.1.** *During GRPO, the first-order expectation of token entropy change is given by:*

$$\mathbb{E}_{k\sim\mathbf{p}'}[\Delta H] = -\eta\,\mathrm{Cov}_{k\sim\mathbf{p}'}(A, r(S_* - E_{i\sim\mathbf{p}}[S_i])), \tag{27}$$

where $\mathbf{p}'$ and $\mathbf{p}$ denote the sampling policy's and the current policy model's output distributions at token $t$, respectively.

**Corollary 4.1.** *Within an GRPO training batch, the first-order expectation of entropy change is given by:*

$$\Delta H_{\mathcal{T}_{\mathcal{B}}} = -\eta \operatorname{Cov}_{\mathcal{B}}(A, r(S_* - E_{i \sim \mathbf{p}}[S_i])). \tag{28}$$

# E SUPPLEMENTAL RESULTS OF THE EXPERIMENT

## E.1 CURVES

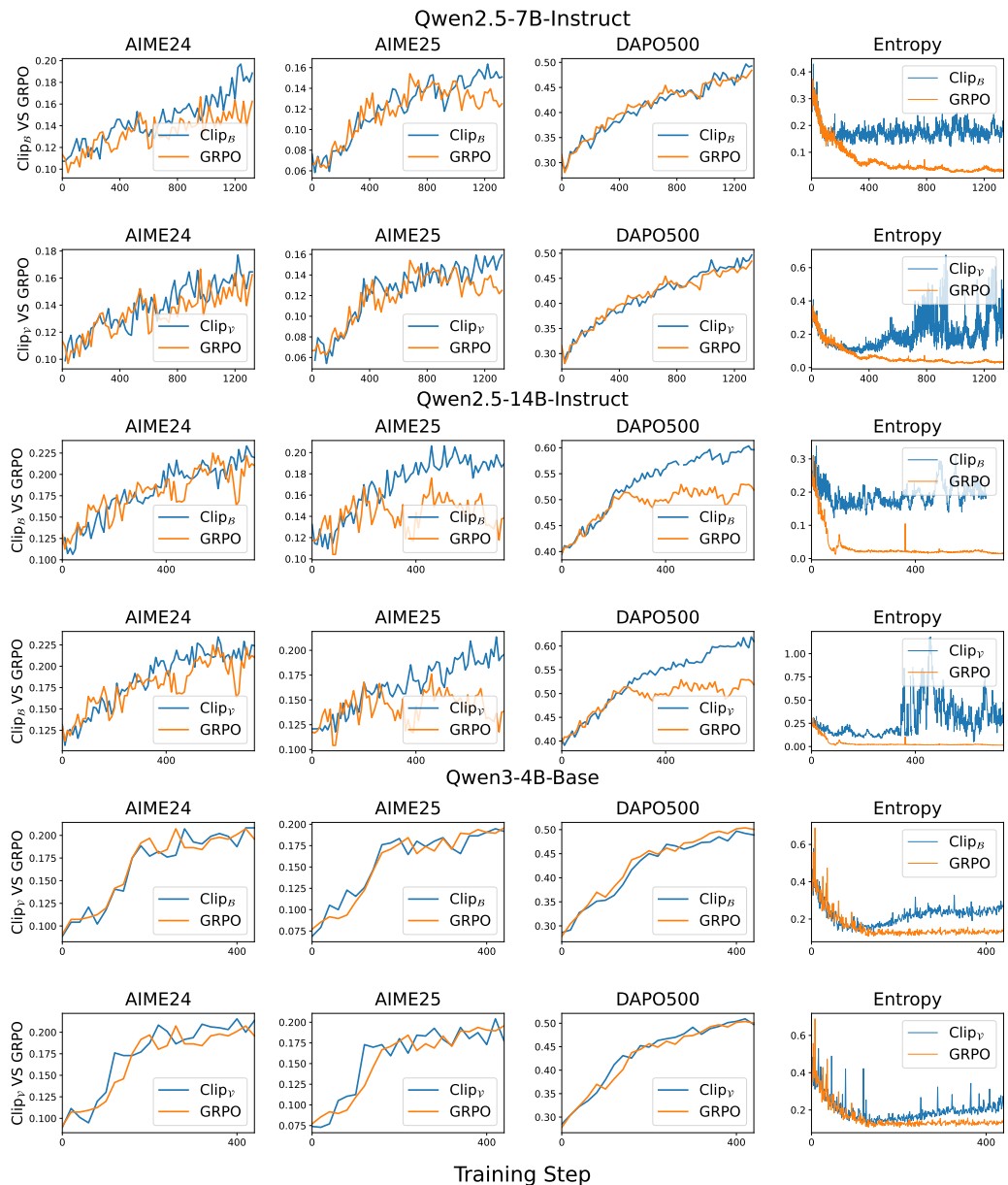

Figure 6: Full curves of performance and entropy for diffirent models.

## E.2 MORE MODELS

We conduct additional experiments on DeepSeek R1-Distill-llama-8B-Instrut (hereafter mentioned as Distilled-Llama), and InternLM3-8B-Instruct (hereafter mentioned as Internlm).

Table 2: Avg@K accuracy of models trained from DeepSeek-Distill-Llama-8B-Instruct.

| Method | AIME24 | AIME25 | DAPO500 |
|---|---|---|---|
| GRPO | 31.87 | 23.85 | 60.12 |
| GRPO+$\text{Clip}_\mathcal{B}$ | 32.19 | 24.37 | **60.42** |
| GRPO+$\text{Clip}_\mathcal{V}$ | **32.40** | **24.69** | 59.88 |

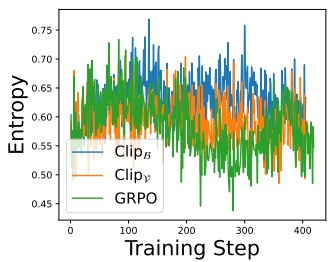

Figure 7: Entropy of Disitilled-Llama during RFT.

For Distilled-Llama, although the model's training dynamics differ significantly from those of Qwen, our method still demonstrates strong entropy-stabilizing properties (see Figure 7) and achieves competitive model performance (as shown in Table 2).

In the training of InternLM, our method demonstrates useful benefit in stabling training. Despite employing additional data filtering and hyperparameter tuning, InternLM consistently suffers from training collapse when using Vanilla GRPO. In contrast, our method enables stable and sustained training. The corresponding training dynamics are shown in Figure 8: Vanilla GRPO exhibits significant gradient fluctuations in the later stages of training, whereas our method remains relatively stable. This suggests that our filtering of outlier tokens also contributes to training stability.

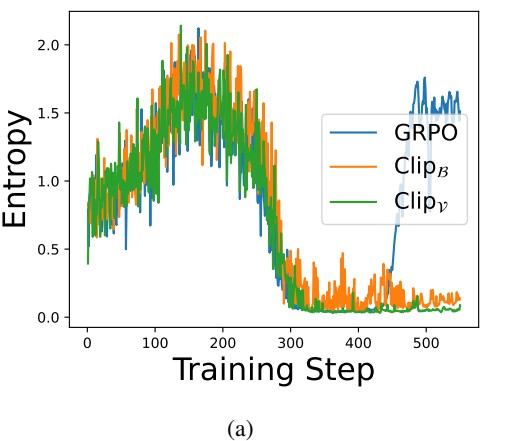

(a)

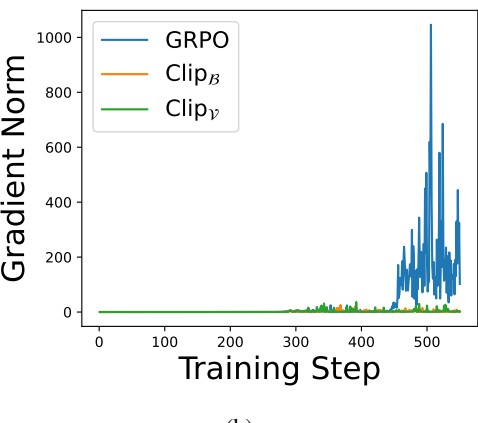

(b)

Figure 8: Dynamics of entropy(a) and gradient norm(b) during RFT of Internlm.

### E.3 MORE BASELINES

We conduct additional experiments (with Qwen2.5-7B-Instruct) to compare $\text{Clip}_\mathcal{B}$ and $\text{Clip}_\mathcal{V}$ with existing practical methods, including clip-cov (Cui et al., 2025), kl-cov (Cui et al., 2025), rewarding the unlikely (He et al., 2025), and entropy regularization (forking tokens) (Wang et al., 2025). The experimental results (evaluated by Avg@k) are shown in Table 3.

These results show that $\text{Clip}_\mathcal{B}$ and $\text{Clip}_\mathcal{V}$, derived from entropy dynamics theory, achieve competitive performance compared to existing practical methods.

### E.4 EXPERIMENTS WITH PPO

we provide a simple demonstration with PPO. We directly apply the GAE Advantage from PPO as the criterion for determining the token optimization direction in our algorithms, i.e.

$$\delta_t = r_t + \gamma V_{t+1} - V_t,$$

Table 3: Comparison between $\text{Clip}_\mathcal{B}/\text{Clip}_\mathcal{V}$ and existing practical methods.

| Method | AIME24 | AIME25 | DAPO500 |
|---|---|---|---|
| Vanilla GRPO | 16.88 | 15.42 | 48.03 |
| Clip-Cov | 20.00 | 16.67 | 48.03 |
| KL-Cov | 20.83 | 15.83 | 48.20 |
| Rewarding unlikely | 15.94 | 15.83 | 48.22 |
| Forking tokens | 16.35 | 14.90 | 47.13 |
| $\text{Clip}_\mathcal{B}$ | 19.69 | 16.35 | 49.68 |
| $\text{Clip}_\mathcal{V}$ | 18.12 | 15.94 | 49.65 |

$$A_t = \delta_t + (\gamma\lambda)A_{t+1} \,,$$

where $V$ denotes the state value assigned by critic model, $\gamma$ denotes the discount factor and $\lambda$ represents smoothing parameter. As a simple and direct application to PPO, our methods achieve significant imporvements, as shown in table 4

Table 4: Experiment results of $\text{Clip}_\mathcal{B}/\text{Clip}_\mathcal{V}$ with PPO training algorithm.

| Method | AIME24 | AIME25 | DAPO500 |
|---|---|---|---|
| Vanilla PPO | 16.15 | 13.75 | 40.98 |
| $\text{Clip}_\mathcal{B}$ | 16.56 | **15.31** | **46.12** |
| $\text{Clip}_\mathcal{V}$ | **17.60** | 15.21 | 44.50 |

We believe this result convincingly demonstrates the potential of our work to be applied across various policy gradient methods, and highlights that developing entropy control methods tailored for different RFT algorithms based on the entropy dynamics is a promising direction for future work.

# F   USAGE OF LARGE LANGUAGE MODELS

We use large language models, including Gemini-2.5-Pro and Qwen3-235B-A22B, to check for grammar errors and proofread this manuscript.

