# OpenReview forum: "On the Entropy Dynamics in Reinforcement Fine-Tuning of Large Language Models"
_ICLR.cc/2026/Conference — Submitted to ICLR 2026_

### Official Review · Reviewer_STqk · 2025-10-29

**Soundness:** 2
**Presentation:** 3
**Contribution:** 2
**Rating:** 4
**Confidence:** 4

**Summary:**

This paper studies entropy dynamics (i.e., the change of policy entropy) in policy gradient methods. It presents a preliminary theoretical analysis of entropy change from the logits perspective. Based on this analysis, the paper proposes entropy-controlling methods that identify tokens within a training batch exerting disproportionate impact on entropy changes. This enables selective mitigation of outlier token influence, achieving fine-grained and flexible control over entropy regularization throughout training.

**Strengths:**

- The theoretical analysis in Lemma 1 and Theorem 2 is clear and well-presented. While the derivations are straightforward, the insights they provide are valuable. In particular, the measurement approach in Equation (6) is both insightful and practically useful.
- The paper provides extensive discussion of existing approaches that fall under this framework, offering a unified view of entropy control methods.

**Weaknesses:**

- The theoretical analysis has notable limitations. Following Ren & Sutherland (2025), the analysis is based on gradients computed from fixed data. This differs from the standard RL setting, where data is sampled from the evolving model distribution, potentially limiting the applicability of the theoretical insights.

- Despite the solid theoretical analysis, the empirical advantage over existing methods remains unclear. Specifically, the paper lacks sufficient comparison with direct entropy regularization methods and the regularization approaches proposed in Cui et al. (2025), Yu et al. (2025), and Wang et al. (2025).

- The empirical improvements over GRPO in Table 1 are marginal, particularly for the Avg@K metric. I am curious about the training dynamics underlying these results. Could the authors provide training curves showing Avg@K performance and entropy changes throughout training? This would help clarify whether the proposed method achieves quite different dynamics.

**Questions:**

I am unclear about the paper's claim that "the derivation in Cui et al. (2025) relies on idealized assumptions that do not reflect the actual training dynamics of RFT." This criticism appears unfounded. In fact, Cui et al.'s analysis seems more aligned with realistic training scenarios by taking into consideration the expectation over model sampling, which better captures the evolving data distribution in RL settings.

---

> ### Author Response · Authors · 2025-11-23
> **Response to  Reviewer STqk (part 1/2)**
>
> We sincerely appreciate your detailed comments and valuable suggestions! We provide the following responses to address your concerns and answer your questions point by point.
>
> ---
>
> > W1: The theoretical analysis has notable limitations. Following Ren & Sutherland (2025), the analysis is based on gradients computed from fixed data. This differs from the standard RL setting, where data is sampled from the evolving model distribution, potentially limiting the applicability of the theoretical insights.
>
>
> Thank you for your constructive comments. We would like to point out that our analysis has been grounded in data sampled from the evolving model distribution, as you suggested.
>
> In the derivation of Theorem 2, we explicitly base our analysis on the evolving model distribution. Specifically, in the term $S_* = S_k$, the index $k$ is sampled according to an evolving model distribution $\mathbf{p} _ {sample}$, and the term $\mathbb{E} _ {i \sim \mathbf{p}}[S_i]$ explicitly incorporates a correction accounting for model-based sampling.
> In Corollary 1, we compute the expectation of the core expression $(S_* - \mathbb{E} _ {i \sim \mathbf{p}}[S_i])$ under model sampling, and Corollary 2 further extends this result to the batch dimension.
>
> We believe that our theoretical analysis is well aligned with the training dynamics of RFT and remains practical. Thank you again for your valuable comments.
>
> > W2: Despite the solid theoretical analysis, the empirical advantage over existing methods remains unclear. Specifically, the paper lacks sufficient comparison with direct entropy regularization methods and the regularization approaches proposed in Cui et al. (2025), Yu et al. (2025), and Wang et al. (2025).
>
>
> Thank you for your valuable suggestions regarding the baselines.
>
> We conduct additional experiments (with Qwen2.5-7B-Instruct) to compare ClipB/ClipV with existing practical methods, including clip-cov[1], kl-cov[1], rewarding the unlikely[2], and entropy regularization (forking tokens)[3]. The experimental results (evaluated by Avg@k) are shown in the table below.
>
> | Method                    | AIME24 | AIME25 | DAPO500 |
> | ------------------------- | ------ | ------ | ------- |
> | Vanilla GRPO              | 16.88  | 15.42  | 48.03   |
> | Clip-cov                  | 20.00  | 16.67  | 48.03    |
> | KL-cov                    | 20.83  |15.83  | 48.20    |
> | Rewarding Unlikely        |15.94 |15.83 |48.22    |
> | Forking tokens            |16.35 |14.90 |47.13     |
> | $\text{Clip}_\mathcal{B}$ | 19.69  | 16.35  | 49.68   |
> | $\text{Clip}_\mathcal{V}$ | 18.12  | 15.94  | 49.65   |
>
> These results show that ClipB/ClipV, derived from entropy dynamics theory, achieve competitive performance compared to existing practical methods.
>
> We have added the above experimental results and discussions to Appendix E.3 in the revised manuscript. Thank you again for your insightful suggestions to further improve our paper.
>
> > W3: Could the authors provide training curves showing Avg@K performance and entropy changes throughout training? This would help clarify whether the proposed method achieves quite different dynamics.
>
> Thank you for your valuable suggestions.
>
> We have added the Avg@K performance and curves of entropy changes for all models throughout training in Appendix E.1,. The curves show that both ClipB and ClipV effectively mitigate entropy collapse during RFT. As model size increases, the performance gain achieved by encouraging diversity becomes increasingly pronounced, which is consistent with the findings reported in Wang et al. (2025) [3].
>
> Thank you again for your valuable suggestions to further improve our paper.

---

> ### Author Response · Authors · 2025-11-23
> **Response to Reviewer STqk (part 2/2)**
>
> > Q1: I am unclear about the paper's claim that "the derivation in Cui et al. (2025) relies on idealized assumptions that do not reflect the actual training dynamics of RFT." This criticism appears unfounded.
>
> Thank you for your comments. We greatly appreciate the contributions of Cui et al. (2025) in studying the training dynamics of RFT, which have inspired many recent studies (including us) to some extent. We have revised the relevant statements in the revised manuscript (section 6) to highlight these contributions.
>
> Below, we summarize the enhancements and contributions of our study in advancing the research on entropy dynamics with practical RFT training:
>
> (i) Firstly, as mentioned in response to W1, we believe that our analysis is grounded in data sampled from the evolving model distribution, as you suggested, and is well aligned with the training dynamics of RFT and remains practical.
>
> (ii) In Theorem 2, we derive the expression for entropy change directly from the GRPO gradient update rule. We compute expectations from two practical perspectives: model sampling (Corollary 1) and batch-level aggregation (Corollary 2). This approach avoids reliance on undefined advantages, ensuring that our algorithm design is closely aligned with the theoretical derivation.
>
> (iii) In Appendix C.1, we provide a derivation that includes the advantage term, starting from Corollary 1, which facilitates comparison with Cui et al. (2025). We note that this derivation involves advantages for tokens not actually sampled, and therefore remains theoretical. Our expression decouples the advantage from the model-sampling-dependent quantity $S_* - \mathbb{E}_{i \sim \mathbf{p}}[S_i]$, offering a clearer and more intuitive understanding of entropy collapse in GRPO.
>
> (iv) Furthermore, building upon Corollary 2, we present a complete batch-level expectation of entropy change in Appendix C.2. This provides a more practical view of the underlying mechanism of entropy dynamics and offers deeper insight into the rationale behind our proposed entropy control algorithms.
>
> We have added the above theoretical analysis and discussions into the revised manuscript. Hope these responses address your concerns. Thank you again!
>
> References:
>
> [1]The entropy mechanism of reinforcement learning for reasoning language models.
> [2]Rewarding the unlikely: Lifting grpo beyond distribution sharpening.
> [3]Beyond the 80/20 rule: High-entropy minority tokens drive effective reinforcement learning for llm reasoning.
>
> ---
>
> We have added the above analysis and experimental results to the revised manuscript. Thank you again for your constructive suggestions to improve the quality of our work. Hope these responses can well address your concerns and lean you more towards the acceptance of our work.

---

> ### Author Response · Authors · 2025-11-28
> **Thank you and look forward to further discussion**
>
> Dear Reviewer STqk:
>
> Thank you for your detailed comments and helpful suggestions! We are wondering if our responses have resolved your concerns. Please let us know if our response and clarification meet your expectations. We are eager to engage in further discussions and continue improving our work.
>
> Best regards,
>
> Authors

---

### Official Review · Reviewer_myqB · 2025-10-31

**Soundness:** 3
**Presentation:** 3
**Contribution:** 2
**Rating:** 6
**Confidence:** 3

**Summary:**

The paper studies how the entropy of large language models changes during reinforcement fine-tuning (RFT). While many prior methods adjust entropy heuristically, this paper builds a theoretical framework to explain these changes in a principled way.

The authors first analyze how updating a single token’s logit affects model entropy and derive a formula linking entropy change to a new “entropy discriminator” score that depends on the token’s probability and overall entropy. They then extend the analysis to Group Relative Policy Optimization (GRPO) and show that entropy change depends on how a token’s discriminator score compares to its policy-wide average. Using these insights, they design two simple entropy control methods—ClipB and ClipV—which clip or mask updates that would cause large entropy shifts.

Experiments on math reasoning datasets (AIME24, AIME25, and DAPO500) show that these methods prevent entropy collapse, improve exploration, and boost Pass@K accuracy on Qwen models.

**Strengths:**

1. This paper provides a clear, simple theoretical insight on the entropy dynamics in the RFT process. The paper derives a compact first-order expression that links a single-logit update to entropy change. This gives an intuitive discriminator that predicts whether an update increases or decreases entropy. The derivation is straightforward and easy to follow (Lemma 1 / Theorem 1). The authors extend the single-token result to a GRPO optimization step. This links per-token effects to the actual optimization algorithm used in practice, improving applicability beyond toy analysis.
2. The analysis provides a single lens to interpret several previously proposed entropy heuristics (clipping schemes, entropy regularization, probability-weighted updates), which is valuable for the community and may reduce ad-hoc tuning.

**Weaknesses:**

1. Corollary 1 appears to be incorrect. It claims that the expected entropy change within GRPO optimization is zero. If I understand correctly, this is proved by calculating the expectation of the first-order entropy change $\Delta H$ under the token distribution $k \sim p$. However, $\Delta H$ contains the advantage $A$, which depends on the token $k$. The paper seems to ignore this dependence.
2. The scope of the theoretical analysis is limited. The analysis is first-order (small perturbation) and studies a single-token update effect. While useful intuitively, RFT involves updates based on responses consisting of multiple tokens. Besides, the paper assumes that many practical algorithmic components including clipped importance sampling ratio, KL penalties and entropy bonuses are inactive; this significantly limits the scope of the analysis.
3. The paper uses Theorem 1 to interpret clipping mechanisms and entropy regularization techniques from previous studies. However, Theorem 1 is overly idealistic—it analyzes updates to a single token's logit. In actual RFT procedures, multiple tokens' logits change within one gradient update. This creates a notable gap between theory and practice.
4. The paper compares against “vanilla GRPO” and interprets prior methods conceptually, but empirical head-to-head comparisons against recent strong entropy-aware methods (e.g., DAPO) are missing. A direct experimental comparison would clarify where ClipB/ClipV stand relative to the best existing practical methods.

**Questions:**

Please refer to the weakness part.

---

> ### Author Response · Authors · 2025-11-23
> **Response to Reviewer myqB (part 1/3)**
>
> We sincerely appreciate your detailed comments and valuable suggestions! We provide the following responses to address your concerns and answer your questions point by point.
>
> ---
>
> > W1: Corollary 1 appears to be incorrect. It claims that the expected entropy change $\Delta H$ within GRPO optimization is zero. If I understand correctly, this is proved by calculating the expectation of the first-order entropy change under the token distribution $k\sim p$. However, $\Delta H$ contains the advantage , which depends on the token . The paper seems to ignore this dependence.
>
> Thank you for your helpful comments. We have revised the statement in Corollary 1 to take sampling into account:
> To a first-order approximation, under on-policy sampling, the expected entropy change factor $\mathbb{E} _ {k \sim \mathbf{p}}[S_*- \mathbb{E} _ {i \sim \mathbf{p}}[S_i]]$ for a token within GRPO optimization is zero.
>
> Besides, we also provide extensions of Corollary 1 to include the suggested advantage term, and calculate corresponding expected values of entropy change. The analysis have been summarized as Corollary 3 and updated in section 3.2. It is worth noting that, in practice, due to the nature of the GRPO algorithm, the advantage is undefined and thus incalculable for tokens that do not appear in the actual sampled trajectories. We therefore focus on the terms in Corollary 1 when designing entropy control algorithms.
>
> The above modification has been updated to section 3.2 in the revised manuscript. Please refer to Appendix C.1 in the revised manuscript for more details. Thank you again for your helpful comments.
>
>
> > W2.1：The scope of the theoretical analysis is limited. The analysis is first-order (small perturbation) and studies a single-token update effect. While useful intuitively, RFT involves updates based on responses consisting of multiple tokens.
>
> Thank you for your constructive comments.
>
> Built upon previous studies[1][2][3][4] that conduct empirical investigations into entropy and probability within rollout tokens, we establish a theoretical framework for analyzing entropy dynamics by starting from a similar setting. Within our first-order approximation theoretical framework, batch-level entropy is typically defined as the arithmetic average of entropies across individual tokens, and the dynamics of batch-level entropy emerge as a linear superposition of token-level entropy dynamics.
>
> According to your suggestions, we provide a batch-level analysis to make the entropy dynamics in this scenario clearer.
> Specifically, the batch-level entropy is defined by the arithmetic average of token entropies within a batch:
>
> $ H_{\mathcal{T_B}} = \frac{1}{|\mathcal{T_B}|}\sum_{t\in\mathcal{T_B}}H^t $
>
> Therefore，batch-level entropy change is also the arithmetic average of token entropy change:
>
> $ \Delta H_{\mathcal{T_B}} = \frac{1}{|\mathcal{T_B}|}\sum_{t\in\mathcal{T_B}} \Delta H^t $
>
> Applying theorem 2 (refer to Eq.9 in the paper), we have:
>
> $ \Delta H_{\mathcal{T_B}} = -\frac{1}{|\mathcal{T_B}|}\sum_{t\in\mathcal{T_B}} \alpha(S_*-E_{i\sim P_t}[S_i^t])] = -\mathbb{E} _ \mathcal{B}[ \alpha(S_*-E_{i\sim P}[S_i]) ], $
>
> where $\mathbb{E}_\mathcal{B}$ refers to statistical expectation. i.e., arithmetic average.
>
> Considering the three factors in $ \alpha $: The learning rate $ \eta $ is constant within a batch; $ r $ is constantly 1 in the on-policy setting. The advantage $ A $estimated in the GRPO algorithm is NOT independent to the chosen token id in a batch $ \mathcal{T_B} $, i.e.,
>
> $ \Delta H_{\mathcal{T_B}} = -\eta\mathbb{E} _ \mathcal{B}[A(S_*-E_{i\sim P}[S_i])] , $
>
> We further apply covariance decomposition to $ \Delta H_\mathcal{T_B} $ within a training batch:
>
> $ \Delta H _ {\mathcal{T_B}}/ \eta
> =-\\{ \mathbb{E} _ {\mathcal{B}}[A]\mathbb  E_{\mathcal{B}}[S_*]
> +\mathrm{Cov} _ {\mathcal{B}}(A,S_*)
> -\mathbb{E} _ {\mathcal{B}}[A]\mathbb  E _ {\mathcal{B}}[E_{i\sim P}[S_i]]
> -\mathrm{Cov} _ {\mathcal{B}}(A,E_{i\sim P}[S_i])\\},
>  $
>
> According to corollary 2 (refer to Eq.11 in the paper), $ \mathbb{E} _ {\mathcal{B}}[S]-\mathbb{E} _ {\mathcal{B}}[E_{i\sim P}[S_i]]=0 $, we have:
>
> $ \Delta H _ {\mathcal{T_B}}/\eta = -\mathrm{Cov} _ {\mathcal{B}}(A,S_*-E_{i\sim P}[S_i]). $
>
> From the experimental results shown in Figure 5 in Appendix C.2, we observe that the value of $ -\mathrm{Cov} _ {\mathcal{B}}(A, S_* - \mathbb{E} _ {i \sim P}[S_i]) $ is predominantly negative. In conjunction with Theorem 2, these observations further confirm that the model tends to obtain correct answers (i.e., $A > 0$) by producing “safe” responses (i.e., those with relatively high probabilities, for which $S - \mathbb{E}[S]$ is positive), whereas exploratory behaviors are more likely to yield incorrect answers. This dynamic persistently suppresses the model’s propensity to explore diverse answers.

---

> ### Author Response · Authors · 2025-11-23
> **Response to Reviewer myqB (part 2/3)**
>
> (Continuing response to W2.1)
>
> Algorithm 2 directly computes the factor $S_* - \mathbb{E} _ {i \sim P}[S_i]$, and masks tokens which contribute extremely significantly to the covariance expression. For example, for negative samples where $A < 0$, Theorem 2 masks tokens with large negative $S_* - \mathbb{E} _ {i \sim P}[S_i]$, which contribute large negative factors, thereby stabilizing the change in entropy. Algorithm 1 estimates this factor from a batch perspective for better computational efficiency.
>
> As a result, the proposed analysis and algorithms can be effectively applied to batch-wise RFT training. We have added the above theoretical analysis, discussions, and additional experimental observations to Appendix C.2 in the revised manuscript. Thank you again for your insightful suggestions to further improve our paper.
>
>
> > W2.2: Besides, the paper assumes that many practical algorithmic components including clipped importance sampling ratio, KL penalties and entropy bonuses are inactive; this significantly limits the scope of the analysis.
>
> Thank you for your thoughtful comments. We have provided point-by-point responses below.
>
> (i) *Clipped importance sampling ratio*: Within our theoretical framework, the gradients of those tokens trigger the clipping mechanism are set to zero, contributing no changes to the entropy. Therefore, we choose to ignore these tokens. We have revised the statements in section 3.1 to make it clearer.
>
> (ii) *KL penalties and entropy bonuses*: Previous studies[1][4][5] indicate that KL penalties and explicit entropy bonuses may negatively impact RFT performance. More recent works[3][6][7] advocate omitting these terms, making their exclusion a practical choice in RFT. Besides, incorporating these auxiliary components would add unnecessary complexity and potentially obscure the core dynamics under investigation. Therefore, we opt for a clear setting to better isolate and elucidate the key mechanisms of the RFT process.
>
> We have revised the relevant statements in Section 3.1 in the revised manuscript to make them clearer. Thank you again for your thoughtful comments.
>
>
> > W3: The paper uses Theorem 1 to interpret clipping mechanisms and entropy regularization techniques from previous studies. However, Theorem 1 is overly idealistic—it analyzes updates to a single token's logit.
>
> Thank you for your comments. We would like to respond from two perspectives.
>
> (i) *Rationale for using Theorem 1*: In Section 4.2, we provide a directional understanding of existing entropy control methods from the perspective of entropy dynamics theory, rather than a rigorous theoretical account. As these methods were not specifically designed with considerations of entropy dynamics in mind, providing a strict theoretical explanation without relying on numerical computation is intractable.
> Although Theorem 1 is derived from a simplified scenario, it has straightforward and readable formulation, and provides directional insights align well with those of Theorem 2, which we hope aids comprehension.
>
>
> (ii) *Analysis based on Theorem 2*: We provide a qualitative analysis based on the more practical Theorem 2:
>
> Firstly, recall Theorem 2 and examine, from a statistical perspective, the relationship it reveals between two factors often considered by the methods above, i.e., probability and entropy.
> The first term in Theorem 2, $S_* = p_k(H+\mathrm{log}\,p_k)$, directly associates these two factors. For tokens sampled with high probability, $S_*$ tends to be larger; similarly, tokens sampled in positions with high entropy also have larger $S_*$. Tokens sampled with larger $S_*$ are more likely to obtain a positive value when calculating the deviation from the expectation $\mathbb{E}_{i\sim k}[S_i]$, as the expectation represents an average value under the current model's sampling distribution.
>
> As a result, when considering changes in entropy $\Delta H$, for positive samples, higher probability and lower token entropy are often associated with a decrease in entropy. In contrast, lower probability and higher entropy are often linked to an increase in entropy. For negative samples, the trend is reversed.
>
> Building upon the analysis above, we can conclude some insights for existing entropy control methods: (a) DAPO reduces clipping on low-probability positive samples (which are associated with entropy increase); (b) Entropy regularization only updates high-entropy tokens (among which positive samples contribute to entropy increase); (c) And probability reweighted updating increases the weight of low-probability positive samples, likewise leading to an increase in entropy.
>
> We have revised the relevant statements in Section 4.2 with more detailed analysis in the revised manuscript to make them clearer. Thank you again for your thoughtful comments.

---

> ### Author Response · Authors · 2025-11-23
> **Response to Reviewer myqB (part 3/3)**
>
> > W4: A direct experimental comparison would clarify where ClipB/ClipV stand relative to the best existing practical methods.
>
>
> Thank you for your valuable suggestions regarding the experimental comparisons.
>
> - Since our analysis and experiments are built within the on-policy setting, it is unsuitable and unfair to compare directly with DAPO designed for off-policy scenarios.
> - We conduct additional experiments (with Qwen2.5-7B-Instruct) to compare ClipB/ClipV with existing practical methods, including clip-cov[4], kl-cov[4], rewarding the unlikely[2], and entropy regularization (forking tokens)[3]. The experimental results (evaluated by Avg@k) are shown in the table below.
>
> | Method                    | AIME24 | AIME25 | DAPO500 |
> | ------------------------- | ------ | ------ | ------- |
> | Vanilla GRPO              | 16.88  | 15.42  | 48.03   |
> | Clip-cov                  | 20.00  | 16.67  | 48.03    |
> | KL-cov                    | 20.83  |15.83  | 48.20    |
> | Rewarding Unlikely        |15.94        |15.83        |48.22         |
> | Forking tokens            |16.35        |14.90        |47.13         |
> | $\text{Clip}_\mathcal{B}$ | 19.69  | 16.35  | 49.68   |
> | $\text{Clip}_\mathcal{V}$ | 18.12  | 15.94  | 49.65   |
>
> These results show that ClipB/ClipV, derived from entropy dynamics theory, achieve competitive performance compared to existing practical methods.
>
> We have added the above experimental results and discussions to Appendix E.3 in the revised manuscript. Thank you again for your insightful suggestions to further improve our paper.
>
> References:
>
> [1]Dapo: An open-source llm reinforcement learning system at scale.
> [2]Rewarding the unlikely: Lifting grpo beyond distribution sharpening.
> [3]Beyond the 80/20 rule: High-entropy minority tokens drive effective reinforcement learning for llm reasoning.
> [4]The entropy mechanism of reinforcement learning for reasoning language models.
> [5]Enhancing efficiency and exploration in reinforcement learning for llms.
> [6]POLARIS: A Post-Training Recipe for Scaling Reinforcement Learning on Advanced Reasoning Models.
> [7]Part i: Tricks or traps? a deep dive into rl for llm reasoning.
>
> ---
>
> We have added the above analysis and experimental results to the revised manuscript. Thank you again for your constructive suggestions to improve the quality of our work. Hope these responses can well address your concerns and lean you more towards the acceptance of our work.

---

> ### Author Response · Authors · 2025-11-28
> **Thank you and look forward to further discussion**
>
> Dear Reviewer myqB:
>
> Thank you for your detailed comments and helpful suggestions! We are wondering if our responses have resolved your concerns. Please let us know if our response and clarification meet your expectations. We are eager to engage in further discussions and continue improving our work.
>
> Best regards,
>
> Authors

---

### Official Review · Reviewer_2wia · 2025-11-01

**Soundness:** 2
**Presentation:** 2
**Contribution:** 2
**Rating:** 6
**Confidence:** 4

**Summary:**

The paper develops a theoretical framework to explain how the entropy of LLMs evolves during reinforcement fine-tuning. The authors start from single-token logit updates and derive a first-order analytical expression for entropy change, showing that it depends on both the update direction and a derived entropy discriminator $S^*$. They extend this analysis to GRPO, introducing an expected entropy baseline that governs whether updates increase or decrease entropy. Based on this framework, the paper proposes two entropy-control methods, ClipB and ClipV, that clip gradients associated with tokens contributing excessively to entropy fluctuations. Empirical experiments on reasoning benchmarks such as AIME24, AIME25, and DAPO demonstrate that these methods mitigate entropy collapse, preserve exploration, and improve problem-solving diversity in RFT-trained LLMs

**Strengths:**

1. This paper’s key strength is to provide a very clean expression of the change of entropy during policy updates.

2. This expression motivates clean and simple clipping techniques (ClipB and ClipV) to improve exploration without destabilizing training.

Overall, the work contributes both theoretical depth and practical utility, marking a valuable advance in understanding and stabilizing reinforcement fine-tuning for LLMs

**Weaknesses:**

1. From an empirical perspective, the experiments are limited to the Qwen model family, leaving uncertainty regarding the generality of the proposed methods across other architectures or training setups. As pointing out in previous paper, Qwen is very different from other models in terms of reasoning.

2. The theoretical analysis presented in Section 3.2 appears broadly applicable to generic policy-gradient algorithms, including not only GRPO but also other methods such as PPO. It is therefore unclear why the proposed entropy-control strategies (ClipB and ClipV) were not evaluated under alternative policy-gradient frameworks.

3. The current analysis is confined to an on-policy setting; an extension or discussion of the off-policy case would significantly strengthen the theoretical completeness of the work.

4. Equation (7) may be confusing at first glance, as it is not immediately evident why the GRPO objective takes that form. In my opinion, it seems more accurate to interpret it as a surrogate loss whose gradient coincides with policy gradient of GRPO loss.

Minor writing suggestion:

1. Few steps in Equation (8) are missed, particularly those involving the gradient of the softmax function, making the result less transparent.

2. Line 161 subscripts and superscripts

**Questions:**

See weakness.

---

> ### Author Response · Authors · 2025-11-23
> **Response to Reviewer 2wia (part 1/2)**
>
> We sincerely appreciate your detailed comments and valuable suggestions! We provide the following responses to address your concerns and answer your questions point by point.
>
> ---
>
> > W1: From an empirical perspective, the experiments are limited to the Qwen model family, leaving uncertainty regarding the generality of the proposed methods across other architectures or training setups.
>
> Thank you for your constructive suggestions. We have conducted additional experiments on DeepSeek R1-Distill-llama-8B-Instruct (hereafter mentioned as Distilled-Llama), and InternLM3-8B-Instruct (hereafter mentioned as Internlm).
>
>
> For Distilled-Llama, although the model's training dynamics differ significantly from those of Qwen, our method still demonstrates strong entropy-stabilizing properties (see Figure 7 in Appendix E.2) and achieves competitive model performance (Avg@K results are shown in the table below).
> |Method|AIME24|AIME25|DAPO500|
> | ------------------------- | ------ | ------ | ------- |
> |GRPO|31.87|23.85|60.12|
> |ClipB|32.19|24.37|60.42|
> |ClipV|32.40|24.69|59.88|
>
> In the training of InternLM, our method demonstrates useful benefit in stablizingng training. Despite employing additional data filtering and hyperparameter tuning, InternLM consistently suffers from training collapse when using Vanilla GRPO. In contrast, our method enables stable and sustained training. The corresponding training dynamics are shown in Figure 8 in Appendix E.2: Vanilla GRPO exhibits significant gradient fluctuations in the later stages of training, whereas our method remains relatively stable. This suggests that our filtering of outlier tokens also contributes to training stability.
>
>
> These experimental results demonstrates that the proposed methods are effective across various model architectures. We have added the above experimental results to Appendix E.2 in the revised manuscript. Thank you again for your suggestions to further improve our paper.
>
>
> > W2: It is therefore unclear why the proposed entropy-control strategies (ClipB and ClipV) were not evaluated under alternative policy-gradient frameworks.
>
> Thank you for your valuable suggestions.
> In our study, the proposed theoretical framework is motivated by the observed entropy collapse issue in GRPO, and we also utilize the GRPO loss function in our analysis (Theorem 2). Therefore, GRPO is primarily adopted as the policy-gradient algorithm in our experiments to provide corresponding empirical observations for the theoretical framework.
>
> Although the derivation of Theorem 2 can be applied to other policy gradient methods (e.g., PPO), these methods differ significantly in how they compute the advantage. As a result, when using Theorem 2 to derive subsequent corollaries and to design entropy control methods, method-specific considerations (such as the influence of the critic model) should be involved. We believe that extending the entropy dynamics framework to a broader range of policy gradient methods is a promising direction for future research.
>
> > W3: The current analysis is confined to an on-policy setting; an extension or discussion of the off-policy case would significantly strengthen the theoretical completeness of the work.
>
> Thank you for your insightful suggestion of extending the analysis to the off-policy setting.
>
> The derivation of Theorem 2 is based on the general GRPO formulation and is not restricted to the on-policy setting. We have extended Corollary 1, Corollary 2, as well as the newly added Corollary 3 and Corollary 4 to their off-policy counterparts, along with the corresponding analysis in the Appendix D of revised manuscript. Here is an example illustrating the extension of Corollary 1.
>
> **Corollary 1.1. (off-policy extension of Corollary 1)**
> To a first-order approximation, the expected entropy change factor $r(S_* - \mathbb{E} _ {i \sim \mathbf{p}}[S_i])$ of a token within GRPO optimization is zero, i.e.,
> $
> \mathbb{E} _ {k \sim \mathbf{p}'}\left[r(S_* - \mathbb{E} _ {i \sim \mathbf{p}}[S_i])\right] = 0.
> $
>
> *Proof.* We derive the results as follows:
>
> $
> \begin{align}
> \mathbb{E} _ {k \sim \mathbf{p}'}[r(S _ * - \mathbb{E} _ {i \sim \mathbf{p}}[S_i])] &=\mathbb{E} _ {k\sim \mathbf{p}'}\\{r[p_k(H+\log p_k)-\sum _ {i=1}^{V}p_i^{2}(H+\log p_i)]\\}\\\\
> &=\sum _ {k=1}^V\frac{p_k}{p'_k}p'_k p_k(H+\log p_k)-\sum _ {i=1}^{V}p_i^{2}(H+\log p_i)\sum _ {k=1}^V\frac{p_k}{p'_k}p_k'\\\\
> &= (1-\sum _ {k=1}^Vp_k)\sum _ {i=1}^{V}p_i^{2}(H+\log p_i) = 0.
> \end{align}
> $
>
> For further details on the extension to the off-policy setting, please refer to Appendix D.

---

> ### Author Response · Authors · 2025-11-23
> **Response to Reviewer 2wia (part 2/2)**
>
> > W4: Equation (7) may be confusing at first glance, as it is not immediately evident why the GRPO objective takes that form. In my opinion, it seems more accurate to interpret it as a surrogate loss whose gradient coincides with policy gradient of GRPO loss.
>
> Thank you for your valuable suggestions! We have revised the expressions related to Equation 7 to make them clearer according to your suggestion, as shown below.
>
> Recall the training objective function of GRPO in equation 1. For a chosen token $a^k$ with token id $k$, its contribution to the whole training target can be expressed as:
> $
> \frac{\mathbf{p} _ k}{p'_ {k}}\cdot A \,
> $
> where $\mathbf{p}_k$ denotes the current model distribution in the sampled position and $p'_k$ is the probability of the sampled token under the sampling model distribution. Therefore, its contribution to the GRPO training loss can be given by a surrogate loss:
>
> \\[
>     \mathcal{L}(\mathbf{z}) = r \cdot A \cdot \log p_k(\mathbf{z}),
> \\]
>
> where $A$ represents the (group-relative) advantage and $r=\pi_\theta(a^k)/\pi_{\theta_{\text{sample}}}(a^k)$ is the importance sampling ratio.
>
> The above modification has been updated in section 3.2 in the revised manuscript. Thank you again for your insightful suggestions to further improve our paper.
>
> > Minor writing suggestion: 1. Few steps in Equation (8) are missed, particularly those involving the gradient of the softmax function, making the result less transparent. 2. Line 161 subscripts and superscripts.
>
> Thank you for your suggestions! We have revised these contents in section 3.1 and 3.2 accordingly, and also further polished the entire manuscript. Thank you again!
>
> ---
>
> We have added the above analysis and experimental results to the revised manuscript. Thank you again for your constructive suggestions to improve the quality of our work. Hope these responses can well address your concerns and lean you more towards the acceptance of our work.

---

> ### Author Response · Authors · 2025-11-28
> **Thank you and look forward to further discussion**
>
> Dear Reviewer 2wia:
>
> Thank you for your detailed comments and helpful suggestions! We are wondering if our responses have resolved your concerns. Please let us know if our response and clarification meet your expectations. We are eager to engage in further discussions and continue improving our work.
>
> Best regards,
>
> Authors

---

> ### Author Response · Authors · 2025-11-28
> **Update: additional experiments with PPO**
>
> To better address your concern regarding the applicability of our method to different policy gradient algorithms, we provide a simple demonstration with PPO. We directly apply the GAE Advantage from PPO as the criterion for determining the token optimization direction in our algorithms, i.e.
> $$
> \delta_t = r_t+\gamma V_{t+1}-V_{t} \\, \,
> $$
>
> $$
> A_t = \delta_t+(\gamma\lambda)A_{t+1} \\, \,
> $$
> where $V$ denotes the state value assigned by critic model, $\gamma$ denotes the discount factor and $\lambda$ represents smoothing parameter.
>
> As a simple and direct application to PPO, our methods achieve significant imporvements, as shown in table below:
>
> | Method                        | AIME24    | AIME25    | DAPO500   |
> | ----------------------------- | --------- | --------- | --------- |
> | Vanilla PPO                   | 16.15     | 13.75     | 40.98     |
> | PPO+$\text{Clip}_\mathcal{B}$ | 16.56 | **15.31** | **46.12** |
> | PPO+$\text{Clip}_\mathcal{V}$ | **17.60**     | 15.21     | 44.50     |
>
> We believe this result convincingly demonstrates the potential of our work to be applied across various policy gradient methods, and highlights that developing entropy control methods tailored for different RFT algorithms based on the entropy dynamics is a promising direction for future work.
>
> ---
> Thank you again for your constructive suggestions to improve the quality of our work. Hope these responses can better address your concerns and lean you more towards the acceptance of our work. We are looking forward to your reply.

---

### Official Review · Reviewer_BPqb · 2025-11-01

**Soundness:** 3
**Presentation:** 3
**Contribution:** 3
**Rating:** 6
**Confidence:** 3

**Summary:**

The article investigates the factors that drive increases and decreases in a model’s output entropy when logits change. By analyzing variations caused by individual tokens and by the GRPO loss function, the authors propose a key metric for computing entropy change. They explain how existing methods affect entropy, how those effects relate to the new metric, and introduce two algorithms built on this metric. Both algorithms also perform strongly on downstream tasks.

**Strengths:**

1. The authors provide a theoretical analysis of entropy variation under changes in logits caused by individual tokens and by gradients derived from the GRPO loss, and they propose a metric for assessing entropy change.

2、 Building on the new metric, the authors establish connections with previous reinforcement learning methods and introduce two new algorithms.

3. Both new algorithms achieve performance improvements on downstream tasks.

**Weaknesses:**

All the analysis is done on "tabular" cases. However, for the RL in LLMs, the updates of different positions and different tokens will be combined by shared parameters in LLM. Why can the theoretical analysis still apply to the scenarios.

**Questions:**

See Weakness.

---

> ### Author Response · Authors · 2025-11-23
> **Response to Reviewer BPqb (part 1/2)**
>
> We sincerely appreciate your detailed comments and valuable suggestions! We provide the following responses to address your concerns and answer your questions point by point.
>
>
> > W1: All the analysis is done on "tabular" cases. However, for the RL in LLMs, the updates of different positions and different tokens will be combined by shared parameters in LLM. Why can the theoretical analysis still apply to the scenarios.
>
> Thank you for your insightful suggestions! We provide the following discussions and theoretical analysis to answer to why the theoretical analysis still applies to the RL in LLMs.
>
> (i) **Why can token-level theoretical analysis be applied to RFT training**:
>
> Built upon previous studies [1][2][3][4] that conduct empirical investigations into entropy and probability within rollout tokens, we establish a theoretical framework for analyzing entropy dynamics by starting from a similar setting. Within our first-order approximation theoretical framework, batch-level entropy is typically defined as the arithmetic average of entropies across individual tokens, and the dynamics of batch-level entropy emerge as a linear superposition of token-level entropy dynamics.
> Specifically, the batch-level entropy is defined by the arithmetic average of token entropies within a batch:
>
> $ H_{\mathcal{T_B}} = \frac{1}{|\mathcal{T_B}|}\sum_{t\in\mathcal{T_B}}H^t $
>
> Therefore, batch-level entropy change is also the arithmetic average of token entropy change:
>
> $ \Delta H_{\mathcal{T_B}} = \frac{1}{|\mathcal{T_B}|}\sum_{t\in\mathcal{T_B}} \Delta H^t $
>
> Applying theorem 2 (refer to Eq.9 in the paper), we have:
>
> $ \Delta H_{\mathcal{T_B}} = -\frac{1}{|\mathcal{T_B}|}\sum_{t\in\mathcal{T_B}} \alpha(S_*-E_{i\sim P_t}[S_i^t])] = -\mathbb{E} _ \mathcal{B}[ \alpha(S_*-E_{i\sim P}[S_i]) ], $
>
> where $\mathbb{E}_\mathcal{B}$ refers to statistical expectation. i.e., arithmetic average.
>
> Considering the three factors in $ \alpha $: The learning rate $ \eta $ is constant within a batch; $ r $ is constantly 1 in the on-policy setting. The advantage $ A $estimated in the GRPO algorithm is NOT independent to the chosen token id in a batch $ \mathcal{T_B} $, i.e.,
>
> $ \Delta H_{\mathcal{T_B}} = -\eta\mathbb{E} _ \mathcal{B}[A(S_*-E_{i\sim P}[S_i])] , $
>
> We further apply covariance decomposition to $ \Delta H_\mathcal{T_B} $ within a training batch:
>
> $ \Delta H _ {\mathcal{T_B}}/ \eta
> =-\\{ \mathbb{E} _ {\mathcal{B}}[A]\mathbb  E_{\mathcal{B}}[S_*]
> +\mathrm{Cov} _ {\mathcal{B}}(A,S_*)
> -\mathbb{E} _ {\mathcal{B}}[A]\mathbb  E _ {\mathcal{B}}[E_{i\sim P}[S_i]]
> -\mathrm{Cov} _ {\mathcal{B}}(A,E_{i\sim P}[S_i])\\},
>  $
>
> According to corollary 2 (refer to Eq.11 in the paper), $ \mathbb{E} _ {\mathcal{B}}[S]-\mathbb{E} _ {\mathcal{B}}[E_{i\sim P}[S_i]]=0 $, we have:
>
> $ \Delta H _ {\mathcal{T_B}}/\eta = -\mathrm{Cov} _ {\mathcal{B}}(A,S_*-E_{i\sim P}[S_i]). $
>
> From the experimental results shown in Figure 5 in Appendix C.2, we observe that the value of $ -\mathrm{Cov} _ {\mathcal{B}}(A, S_* - \mathbb{E} _ {i \sim P}[S_i]) $ is predominantly negative. In conjunction with Theorem 2, these observations further confirm that the model tends to obtain correct answers (i.e., $A > 0$) by producing "safe" responses (i.e., those with relatively high probabilities, for which $S - \mathbb{E}[S]$ is positive), whereas exploratory behaviors are more likely to yield incorrect answers. This dynamic persistently suppresses the model’s propensity to explore diverse answers.
>
> Algorithm 2 directly computes the factor $S_* - \mathbb{E} _ {i \sim P}[S_i]$, and masks tokens which contribute extremely significantly to the covariance expression. For example, for negative samples where $A < 0$, Theorem 2 masks tokens with large negative $S_* - \mathbb{E} _ {i \sim P}[S_i]$, which contribute large negative factors, thereby stabilizing the change in entropy. Algorithm 1 estimates this factor from a batch perspective for better computational efficiency.
>
> As a result, the proposed analysis and algorithms can be effectively applied to batch-wise RFT training. We have added the above theoretical analysis, discussions, and additional experimental observations to Appendix C.2 in the revised manuscript. Thank you again for your insightful suggestions to further improve our paper.

---

> ### Author Response · Authors · 2025-11-23
> **Response to Reviewer BPqb (part 2/2)**
>
> (2) **Regarding shared parameters in LLM**:
>
> When accounting for parameter sharing among different tokens, the theoretical analysis of training dynamics becomes more challenging due to the black-box nature of neural networks. This represents a promising direction for future research, but lies outside the scope of this paper.
> Nevertheless, given the pronounced sparsity observed in activations of LLMs [5][6][7][8], we believe that our theoretical framework and empirical findings can provide useful guidance and insights to the research community for practical training scenarios, as is the case in many token-level analyses and designs discussed in previous studies[1][2][3][4].
>
> Thank you again for your suggestions!
>
> References
> [1]Dapo: An open-source llm reinforcement learning system at scale.
> [2]Rewarding the unlikely: Lifting grpo beyond distribution sharpening.
> [3]Beyond the 80/20 rule: High-entropy minority tokens drive effective reinforcement learning for llm reasoning.
> [4]The entropy mechanism of reinforcement learning for reasoning language models.
> [5]Cats: Contextually-aware thresholding for sparsity in large language models.
> [6]Powerinfer: Fast large language model serving with a consumer-grade gpu.
> [7]Relu strikes back: Exploiting activation sparsity in large language models.
> [8]Training-free activation sparsity in large language models.
>
> ---
>
> We have added the above theoretical analysis, discussions, and experimental results to Appendix C.2 in the revised manuscript. Thank you again for your constructive suggestions to improve the quality of our work. Hope these responses can well address your concerns and lean you more towards the acceptance of our work.

---

> > ### Comment · Reviewer_BPqb · 2025-11-28
> >
> > Thanks for the authors' response.  I have no more questions, and I will keep the score.

---

> > > ### Author Response · Authors · 2025-11-28
> > > **Thank you for your reply**
> > >
> > > Thank you for your response and for acknowledging our submission with positive feedback! We are pleased that our responses have addressed your concerns.
> > >
> > > Thank you again for your efforts in reviewing our paper and providing helpful suggestions!

---

### Author Response · Authors · 2025-11-26
**General responses and look forward to further discussions**

Dear Reviewers,

We hope this message finds you well. We sincerely appreciate your time in providing detailed comments and helpful suggestions! We would like to summarize the comments provided by reviewers and our responses.

---

First of all, reviewers give us positive comments acknowledging that:

- Clear theoretical representation of entropy change during RFT (from Reviewers BPqb, 2wia, myqB, STqk);
- Useful insights for application (from Reviewer myqB, STqk);
- Simple and clean new clipping algorithms (from Reviewers BPqb,2wia);
- Unified lens for understanding entropy control methods (from Reviewer BPqb, myqB, STqk).

---

During the rebuttal period, we have made our best efforts to provide a detailed response to every question from the reviewers for addressing all of their concerns, including:

**Theorical analysis extension**:
- Extending the theorical analysis to the expectation of entropy change under model sampling and batch averaging (please refer to *Appendix C*);
- Extending the theorical analysis to off-policy scenarios (please refer to *Appendix D*);
- Add two further corollaries, Corollary 3 and Corollary 4 (please refer to *Section 3.2*)

**Experiments regarding more baselines and models**:
- Conducting additional experiments regarding more baselines to further confirm the effectiveness of the proposed method (please refer to *Appendix E.3*);
- Conducting additional experiments with more policy models (please refer to *Appendix E.2*).

**Statement enhancement and paper polish for better understanding**:
- Further expanding on the definitions and derivations around Equation 7 and Equation 8 to make them easier to understand (please refer to *Section 3.2*);
- Revising the expressions in Corollary 1 (please refer to *Section 3.2*)
- Using Theorem 2 to explain the entropy control methods (please refer to *Section 4.2*);
- Revising some sentences about related works to better highlight their contributions and clarify our relationship to them (please refer to *Section 6*).

We have uploaded **a revised paper that includes all the experiments, analysis, and discussions** in the above responses. We believe that all the raised questions have been thoroughly addressed and all the misunderstandings have been clarified in our responses. We definitely believe that this submission has been further improved according to your helpful suggestions.

---

We are wondering if our response meets your expectations. We are eager to engage in further discussions and continue improving our work. Thank you again!

---

### Author Response · Authors · 2025-12-03
**Summary of Author Responses and Revisions**

Dear Area Chairs:

Thank you very much for your time and efforts dedicated to our submission! We would like to summarize the comments provided by the reviewers and our corresponding responses and improvements in the revised paper.

> **Positive Comments from Reviewers**

First of all, the reviewers gave us positive comments acknowledging:

- Clear theoretical representation of entropy change during RFT (from Reviewers *BPqb, 2wia, myqB, STqk*);
- Useful insights for application (from Reviewers *myqB, STqk*);
- Simple and clean new clipping algorithms (from Reviewers *BPqb, 2wia*);
- Unified lens for understanding entropy control methods (from Reviewers *BPqb, myqB, STqk*).

> **Our Responses and Revisions**

During the discussion period, we made our best efforts to provide detailed responses to every question from the reviewers, addressing all of their concerns, including:

**Part A. Theoretical Analysis Extension**

1. We extended our theoretical analysis to the expectation of entropy change under model sampling and batch-level averaging, which have been added to `Section 3.2`, `Appendix C.1`, and `Appendix C.2` in the revised paper.
    (Responses to Reviewer *BPqb-W1*, *myqB-W2*, and *STqk-W1*)

2. We extended the theoretical analysis to off-policy scenarios, which has been added to `Appendix D` in the revised paper.
    (Responses to Reviewer *2wia-W3*)

3. Inspired by the theoretical analysis extension, we added two further corollaries (Corollary 3 and Corollary 4), which have been included in `Section 3.2` in the revised paper.
    (Responses to Reviewer *2wia-W3* and *myqB-W1*)

**Part B. Experiments on More Baselines, Models, and Algorithms**

1. We added the Avg@K performance and curves of entropy changes for all models throughout training, showing that the proposed method effectively mitigates entropy collapse during RFT. These curves have been included in `Appendix E.1` in the revised paper.
    (Responses to Reviewer *STqk-W3*)

2. We conducted additional experiments on DeepSeek R1-Distill-llama-8B-Instruct and InternLM3-8B-Instruct, demonstrating that the proposed methods are effective across various model architectures. These results are presented in `Appendix E.2` in the revised paper.
    (Responses to Reviewer *2wia-W1*)

3. We performed additional comparisons with more baselines (e.g., clip-cov, kl-cov, rewarding the unlikely, and entropy regularization), further confirming the effectiveness of the proposed method. These results have been added to `Appendix E.3` in the revised paper.
    (Responses to Reviewer *myqB-W4* and *STqk-W2*)

4. We conducted further experiments with PPO as the policy gradient algorithm, showing the potential for our work to be applied across different policy gradient algorithms. These results have been included in `Appendix E.4` in the revised paper.
    (Responses to Reviewer *2wia-W2*)

**Part C. Statement Enhancement and Paper Polishing for Better Understanding**

1. We further expanded definitions and derivations around Equation (7) and (8) to make them clearer. These revisions have been added to `Section 3.2`.
    (Responses to Reviewer *2wia-W4 and minor writing suggestions*)

2. We revised the expressions in Corollary 1 for better understanding. These revisions are now in `Section 3.2`.
    (Responses to Reviewer *myqB-W1*)

3. We adopted Theorem 2 to bridge entropy dynamics to entropy control methods, making it more practical. These revisions have been incorporated into `Section 4.2`.
    (Responses to Reviewer *myqB-W3*)

4. We summarized the contributions of some existing works, and highlighted the improvements of our study in advancing research on entropy dynamics with practical RFT training. These updates appear in `Section 6`.
    (Responses to Reviewer *STqk-Q1*)

---

We have uploaded **a revised paper that includes all the experiments, analysis, and discussions** in our responses. We believe that all the raised questions have been thoroughly addressed and all misunderstandings clarified.

Thank you again for your time and efforts!

---

### Meta-Review · Area_Chair_7mwe · 2026-01-02

**Summary:**

Reviewers generally found the token-level analysis of entropy change to be clear and conceptually interesting, and appreciated the attempt to connect it to entropy-control mechanisms in reinforcement fine-tuning. However, the decision-relevant concerns centered on the limited realism and scope of the theoretical framework, which relies on first-order, token-level assumptions that do not fully capture coupled multi-token updates and parameter sharing in large language models. In addition, reviewers expressed uncertainty about the practical impact of the proposed methods, noting that empirical gains over existing entropy-control approaches were often modest and not consistently convincing. While several reviewers were marginally above the acceptance threshold, these unresolved concerns kept the paper in a borderline range, with one reviewer below the threshold.

**Reviewer Concerns:**

The rebuttal addressed several surface-level concerns by adding experiments on additional model architectures, providing a demonstration under PPO, clarifying parts of the GRPO derivation, and expanding baseline comparisons and training curves. These changes improved clarity and empirical context. However, the core limitations remain: the theoretical analysis continues to rest on idealized first-order and token-level assumptions, and the gap between the theory and realistic RFT dynamics in LLMs remains unaddressed. Moreover, while additional experiments were provided, the evidence does not clearly establish a strong or consistent advantage over existing entropy-control methods. As a result, confidence in the breadth and robustness of the contribution remains limited.

**Reviewer Scores:**

Reviewer BPqb would likely keep their score unchanged, as their primary concern about applicability to LLMs with shared parameters was acknowledged but not resolved. Reviewers 2wia and myqB might view the rebuttal as clarifying and incremental, but would likely keep their scores given the remaining theoretical and empirical limitations. Reviewer STqk, even if they would likely appreciate the added baselines and training curves, would remain below the acceptance threshold due to continued skepticism about realism and impact.

---

### Decision · Program_Chairs · 2026-01-26

Reject